# A global monthly climatology of total alkalinity: a neural network approach

Daniel Broullón[1], Fiz F. Pérez[1], Antón Velo[1], Mario Hoppema[2], Are Olsen[3], Taro Takahashi[4], Robert M. Key[5], Toste Tanhua[6], Melchor González-Dávila[7], Emil Jeansson[8], Alex Kozyr[9] and Steven M.A.C. van Heuven[10]

[1]Instituto de Investigaciones Marinas, CSIC, Eduardo Cabello 6, 36208 Vigo, Spain

[2]Alfred Wegener Institute Helmholtz Centre for Polar and Marine Research, Postfach 120161, 27515 Bremerhaven, Germany

[3]Geophysical Institute, University of Bergen and Bjerknes Centre for Climate Research, Allègaten 70, 5007 Bergen, Norway

[4]Lamont-Doherty Earth Observatory of Columbia University, Palisades, NY 10964, USA

[5]Atmospheric and Oceanic Sciences, Princeton University, 300 Forrestal Road, Sayre Hall, Princeton, NJ 08544, USA

[6]GEOMAR Helmholtz Centre for Ocean Research Kiel, Düsternbrooker Weg 20D-24105 Kiel, Germany

[7]Instituto de Oceanografía y Cambio Global, IOCAG, Universidad de Las Palmas de Gran Canaria, Las Palmas de Gran Canaria, Spain

[8]Uni Research Climate, Bjerknes Centre for Climate Research, Jahnebakken 5, 5007 Bergen, Norway

[9]NOAA National Centers for Environmental Information, 1315 East-West Hwy Silver Spring, MD 20910 USA

[10]Faculty of Science and Engineering, Isotope Research – Energy and Sustainability Research Institute Groningen, University of Groningen, Nijenborgh 6, 9747 AG Groningen, The Netherlands

*Correspondence to*: Daniel Broullón (dbroullon@iim.csic.es)

**Abstract.** Global climatologies of the seawater $CO_2$ chemistry variables are necessary to assess the marine carbon cycle in depth. The climatologies should adequately capture seasonal variability to properly address ocean acidification and similar issues related to the carbon cycle. Total alkalinity ($A_T$) is one variable of the seawater $CO_2$ chemistry system involved in ocean acidification and frequently measured. We used the Global Ocean Data Analysis Project version 2 (GLODAPv2) to extract relationships among the drivers of the $A_T$ variability and $A_T$ concentration using a neural network (NNGv2) to generate a monthly climatology. 99% of the GLODAPv2 dataset used was modelled by the NNGv2 with a root-mean-squared error

(RMSE) of 5.1 µmol kg$^{-1}$. Validation tests with independent datasets revealed the good generalization of the network. Data from five ocean time-series stations showed an acceptable RMSE range of 3.1-6.2 µmol kg$^{-1}$. Successful modeling of the monthly A$_T$ variability in the time-series suggests that the NNGv2 is a good candidate to generate a monthly climatology. The

monthly climatological fields of AT were obtained passing World Ocean Atlas 2013 (WOA13) monthly climatologies through the NNGv2. The spatiotemporal resolution is set by WOA13: 1ºx1º in the horizontal, 102 depth levels (0-5500m) in the vertical, and monthly temporal resolution. The product is distributed through the data repository of the Spanish National Research Council (CSIC; doi: http://dx.doi.org/10.20350/digitalCSIC/8564).

## 1 Introduction

Because of its interaction with the atmospheric carbon dioxide, the marine carbon cycle has fundamental significance for the Earth's climate (Tanhua et al., 2013). The oceanic capacity to dissolve and store atmospheric $CO_2$, and the subsequent chemical speciation, have resulted in approximately 30% less anthropogenic $CO_2$ in the atmosphere (Le Quéré et al., 2017) than it would otherwise have. One unfortunate byproduct of this process is ocean acidification (Doney et al., 2009). As the ocean absorbs anthropogenic $CO_2$, the seawater pH decreases being the main change in the ocean chemistry which defines ocean acidification.

Combined with other climate change effects (e.g., temperature increase and deoxygenation), this process could have severe consequences for marine ecosystems (Orr et al., 2005; Fabry et al., 2008; Hoegh-Guldberg and Bruno, 2010; Kroeker et al., 2013) and, consequently, for life on our planet.

Detailed spatiotemporal knowledge about the marine carbon cycle is necessary to understand and evaluate the consequences of climate change. There are 4 variables of the seawater $CO_2$ chemistry more frequently measured in carbon chemistry

campaigns: total alkalinity (A$_T$), total dissolved inorganic carbon (TCO$_2$, also known as DIC), partial pressure of $CO_2$ (pCO$_2$) and pH. A$_T$ is a key variable in the framework of ocean acidification because of what it is associated: the oceanic capacity to buffer pH changes. Dickson (1981) defined A$_T$ as:

$$A_T = [HCO_3^-] + 2[CO_3^{2-}] + [B(OH)_4^-] + [OH^-] + [HPO_4^{2-}] + 2[PO_4^{3-}] + [SiO(OH)_3^-] + [HS^-] + 2[S^{2-}] + [NH_3]$$

$$- [H^+] - [HSO_4^-] - [HF] - [H_3PO_4] \tag{1}$$

The global A$_T$ distribution is a result of physical and biogeochemical processes that change the concentration of species in Eq. (1) (Wolf-Gladrow et al., 2007). Processes that change salinity are the most influential. The strong linear correlation between salinity and A$_T$ is well documented (e.g. Millero et al., 1998; Friis et al., 2013; Takahashi et al., 2014). In the surface layer precipitation and evaporation are the primary processes that control the A$_T$ distribution. Rivers and submarine groundwater discharge can affect marine A$_T$ locally, with the degree controlled by runoff and the riverine A$_T$ (Hoppema, 1990; Anderson,

2004; Schneider et al., 2007; Cooper et al., 2008). The formation and dissolution of carbonate minerals also contribute to A$_T$ variability (Fry et al., 2015). Upwelling areas that overlie zones of relatively shallow subsurface carbonate dissolution can also

have elevated surface $A_T$ (Millero et al., 1998; Fine et al., 2017). Organic matter cycling can also contribute to $A_T$ changes. This mechanism can be reflected through the consumption and regeneration of nutrients and oxygen (Brewer and Goldman, 1976; Wolf-Gladrow et al., 2007). Finally, hydrothermal vents could modify the concentration of $A_T$ locally (Chen, 2002).

In addition to the spatial variability, most of the drivers mentioned above generate seasonal $A_T$ variability. Phytoplankton blooms (i.e., primary production) and the seasonality in upwelling and river flows are some of the more remarkable processes associated with the time variability of $A_T$. Even though $A_T$ is the variable of the seawater $CO_2$ chemistry system with the least seasonal variability (Lee et al. (2006) estimated a range from near 0 up to 80 µmol kg$^{-1}$), it is important to account for such changes because of the strong connection of $A_T$ with oceanic anthropogenic carbon storage (Renforth and Henderson, 2017)

and to buffer seawater pH changes. A monthly $A_T$ climatology that captures most of the spatiotemporal variability can be used as initial and/or boundary conditions in biogeochemical models, in evaluating the $CaCO_3$ pump (e.g., Carter et al., 2014) or computing the ocean inventory of anthropogenic $CO_2$ (e.g., Steinfieldt et al., 2009).

High-quality data is a crucial first requirement to address the problem. Ocean time-series data represent excellent records to study the seasonality of the ocean carbon cycle as well as its inter-annual trends (e.g., Bates et al., 2014). Unfortunately, there

are only a few time-series that include sufficiently precise measurements of the seawater $CO_2$ chemistry at seasonal resolution. Alternately, various global data products have been released for public usage in recent years. The main ones for the surface ocean are the Surface Ocean $CO_2$ Atlas (SOCAT; Bakker et al., 2016) and the Lamont-Doherty Earth Observatory database (LDEO; Takahashi et al., 2016). These two are complementary, offer annual updates and include tens of millions of $pCO_2$ measurements in the global ocean. For the interior ocean, a comprehensive and global database and data product was recently

made public: Global Ocean Data Analysis Project version 2 (GLODAPv2) (Key et al., 2015; Olsen et al., 2016). This quality-controlled collection contains thousands of measured seawater data, including $CO_2$ chemistry variables, over the full water column from more than 700 globally distributed cruises over the past four decades.

The logical next step is to generate a globally consistent climatology for the different variables that captures seasonal variability. Different approaches have been used to fill spatial and temporal gaps in $A_T$ observations to generate a global

seasonal climatology (Lee et al., 2006; Takahashi et al., 2014). These studies only cover the surface ocean. However, a robust climatology of the entire water column is necessary to assess more than surface ocean.

In this study, we present a global monthly climatology for $A_T$ in a 1°x1° grid in the upper 102 standard depth levels (between 0 and 5500m) of the World Ocean Atlas 2013 (WOA13) designed using a neural network approach. Other studies have demonstrated the capacity of these techniques to reconstruct global $pCO_2$ variability at monthly resolution over the last few

decades (e.g., Landschützer et al., 2013, 2014). Our $A_T$ climatology uses available high-quality measurements and the neural network ability to capture natural variability. We were able to reduce the errors obtained by the previous efforts to build a seasonal $A_T$ climatology (Lee et al., 2006; Takahashi et al., 2014) and to extend the climatology through the water column.

## 2 Methodology

### 2.1 Neural network design

A feed-forward neural network was configured to compute $A_T$ globally at monthly resolution. It was selected based on the ability to learn the relationships between $A_T$ and the variables related to its spatiotemporal variability as shown in Velo et al. (2013).

Feed-forward neural networks are composed of layers: the input layer, a variable number of hidden layers and the output layer (Fig. 1). The input layer is a matrix representing the entry to the network of the data from which the outputs will be obtained.

The hidden and output layers are composed of neurons. The number of these elements in the hidden layers is adjustable and in the output layer is dependent on the number of network outputs. The neurons are formed by a series of weights, a bias, a summation, and a transfer function (Russell and Norvig, 2010). They are the connections between the layers. A neuron receives all outputs from the previous layer and multiplies them by a matrix of weights. These results are summed and a bias is added. Finally, the transfer function is applied over the sum and an output is obtained from each neuron.

The ability of the network to produce a reasonable output stems from a training process. Given a set of inputs and their targets, the network is trained to learn the relationships between both sets. The training process is possible due to a backpropagation training algorithm (Rumelhart et al., 1986). Generally, the network is initialized with random values of weights and biases and an output is obtained. This output is compared with the target through a cost function, that typically is the mean squared error. Then, the algorithm "backpropagates" this error through the network and iteratively adjusts the weights and biases to minimize

the cost function. The minimization is commonly based on the Levenberg-Marquardt algorithm (Levenberg, 1944; Marquardt, 1963). Once the network is trained, output values can be obtained from a set of inputs with unknown targets. The more accurate and generalized the training data, the more accurate the output values.

The feed-forward neural network used in this study has a two-layer architecture. The first layer has a sigmoid transfer function and the second layer a linear transfer function (Fig. 1). This choice of functions allows both the linear and non-linear

relationships between $A_T$ and its predictors to be represented. This network configuration can approximate most functions arbitrarily well (Hagan et al., 2014). In the Atlantic Ocean, this arrangement has been shown to accurately estimate $A_T$ from diverse predictors (Velo et al., 2013).

The GLODAPv2 discrete data were used to train the network. Input variables (left hand in Fig. 1) were selected based on their potential influence on $A_T$ following Velo et al. (2013). They include the sampling position (coordinates and depth), potential

temperature, salinity, nutrients (phosphate, nitrate and silicate) and dissolved oxygen. Position was included to help the network learn characteristic patterns associated with this input when the other variables cannot fully explain the $A_T$ variability Takahashi et al. (2014) and Lee et al. (2006) showed how the relations between $A_T$ and the predictor variables used in these

studies are different depending on the ocean area. The periodicity of the input longitude was represented by the equations used by Zeng et al. (2014):

$$clongitude = \cos\left(\frac{\pi}{180} \cdot longitude\right)$$
(2)

$$slongitude = \sin\left(\frac{\pi}{180} \cdot longitude\right)$$
(3)


Our approach only uses measured inputs from GLODAPv2, that is, those input data derived from the same Rosette sample bottle as the $A_T$ value. Other studies with a similar approach take the inputs from reanalysis products or satellite data (e.g., Landschützer et al. 2013), that are inherently less accurate than direct measurements. The relations created by the network in the training procedure are likely to be more realistic using in situ measured values for the input variables.


The samples where all input variables and $A_T$ were measured were selected from GLODAPv2 (https://www.nodc.noaa.gov/ocads/oceans/GLODAPv2/). From these, we removed one record due to its spurious oxygen value ($O2=1026.9$ µmol kg$^{-1}$ cruise=102; station=4; bottle=5). The final dataset contained 246,221 samples. "GLODAPv2" hereinafter refers to the subset used in this study unless otherwise indicated.


Two different training techniques were tested: the Levenberg-Marquardt method (lm) and the Bayesian Regularization (br) (both detailed in Hagan et al., 2014). In a similar study, Velo et al. (2013) demonstrated that these techniques give the best network performance among those they tested. Except for the number of neurons, the two algorithms were implemented with the default options of the MATLAB functions *trainlm* and *trainbr* (detailed in Beale et al., 2017). These two functions prevent overfitting in different ways. The *trainlm* function usually needs to be fed with the data divided in three sets: a training set to obtain the relationships between variables, a validation set to prevent overfitting and a test set to compare different networks. Here, the training was stopped when the error in the validation set increased during 6 consecutive iterations of the training process to avoid overfitting. This process is known as early stopping (Hagan et al., 2014). The final values of the network weights and biases are those reached before the first of these iterations. The *trainbr* function adds a regularization parameter to the cost function to make the fit smoother in order to avoid overfitting. The validation set is not present in this technique. The end of the training is based on network convergence through parameter stabilization by an automatic process known as automated Bayesian Regularization (Hagan et al., 2014; Beale et al., 2017). See Beale et al. (2017) and references therein for a detailed description of the two functions tested.



The number of network neurons is problem dependent with no fixed criterion for establishment. It is related to the complexity of the input-output mapping, the amount of training data available and their noise (Gardner and Dorling, 1998). Using too few neurons will not enable to learn complex relations. Using too many neurons could overfit the data, that is, the network might


model the uncertainty of the data used in the training. We determined the optimal number of neurons through a trade-off between the root-mean-squared error (RMSE) of the computed values and the generalization of the network. This last concept refers to network performance when a set of unused inputs is passed through the network to obtain an output. If the RMSE in this set is of the same order of magnitude as the RMSE in the training set, there is no substantial overfitting and the network generalizes well.

The training procedure was carried out in MATLAB. We tested 16, 32, 64, 128 and 264 neurons in the hidden layer based on the results of Velo et al. (2013). For each number of neurons, we trained 10 networks always using the same 90% of GLODAPv2 for training (Fig. 2, Static level). The remaining 10% was used as a static test (Fig. 2, Static level). Both subsets contained samples randomly distributed in the ocean to evaluate the maximum possible relationships between the input variables and $A_T$ through all oceanographic regimes, that is, to capture most of the variability in all the variables and not restricting the sets to specific areas. Each of the 10 networks starts the training procedure with random weight and bias values and a random division of the training static dataset into three portions: 70% for training, 15% for testing and 15% for validation (Fig. 2, Dynamic level). These differences make minimization of the cost function different for each network due to the complexity of the weight-error space and, consequently, their different starting points in that space. As each network is different, keeping static sets allows one to determine which network best generalizes in the same test set. The selected network is the one that produces the lowest RMSE in the training data (validation + training dynamic) and in the test data (static + dynamic), considering a non-significant difference between both RMSEs to prevent overfitting. The network derived from this process will be referred as NNGv2.

Once we found an adequate network configuration, we increased the amount of data in the training dynamic set to capture more relations between the inputs and $A_T$. The new percentages of the dynamic sets were: 80% training, 20% validation and 0% testing. The latter set is only necessary to compare different models and is not used during the training. However, the static test set was held to evaluate the generalization of each of the 10 networks to select the best one.

As a last step we eliminated the data points with a difference between measured and computed $A_T$ with the selected network (residuals) beyond ±3RMSE and then retrained the network as above. This procedure was used to identify regions where the network was unable to obtain accurate values and to improve the network mapping in the other areas omitting in this way data that the network could be trying to model without having the appropriate input variables or because they could be data with high measurement errors. Although a well-trained neural network avoids modeling the error, high errors could slightly modify the derived function in a negative manner. The network derived from this process will be referred as NN±3RMSE.

**2.2 Comparison of methods**

The relations proposed by Lee et al. (2006) and Takahashi et al. (2014) to generate a monthly surface climatology of $A_T$ from different predictors were applied over GLODAPv2. Lee et al. (2006) grouped $A_T$ data (< 20-30 m depth) into 5 oceanographic regimes and obtained a best fit to a quadratic function of sea surface temperature (SST) and sea surface salinity (SSS) in each basin. Takahashi et al. (2014) divided the global ocean into 33 hydrographic provinces and expressed the potential alkalinity (PALK = $A_T$ + $NO_3^-$ , < 50 m depth) as a linear regression of salinity in 27 of them. PALK was used instead of $A_T$ for the purpose of eliminating seasonal biological effects, and the inter-province variation reflected differences in $CaCO_3$ production in the mixed layer as well as the contributions of lateral and vertical mixing of waters. The analysis was carried out in the areas defined in the two studies.

The recent methods to compute $A_T$ proposed by Carter et al. (2018) and Bittig et al. (2018) (LIARv2 and CANYON-B respectively) were also compared to the one proposed here. LIARv2 is based on multilinear regressions (MLRs) including the same predictors used in the present study, excluding phosphate (sample position, salinity (S), potential temperature (θ), nitrate (N), apparent oxygen utilization (AOU) and silicate (Si)). This method is composed of 16 equations with a different combination of the input variables, always maintaining the salinity input in each one. The computations with LIARv2 were obtained by the equation with the lowest uncertainty estimate in each sample that this method determines (Carter et al., 2018). CANYON-B is based on a Bayesian neural network derived from GLODAPv2 data including position, time, salinity, temperature and dissolved oxygen as predictors. The two methods were applied on the GLODAPv2 dataset used here and the on a subset excluding the samples where the quality control (QC) of $A_T$ was not done (QC procedures detailed in Olsen et al. (2016) and references therein).

**2.3 Validation**

To illuminate the complexity of neural networks, several methods to determine the contribution of each predictor variable in the output were proposed in different studies (see Gevrey et al. (2003) and Olden et al. (2004)). We used the Connection Weight Approach (Olden and Jackson, 2002) to evaluate if the network properly associates the $A_T$ variability with the predictor variables. This method was proposed to be the most accurate (Olden et al., 2004). It uses the weights obtained in the training stage to extract the influence of each predictor variable in fitting the $A_T$ values. The expression followed was:

$$C_i = \sum_{k=1}^{H} w_{ik} \cdot w_k \tag{4}$$

where $C_i$ is the relative importance of the predictor variable i, H is the number of neurons in the hidden layer, $w_{ik}$ is the weight of the connection between the variable i and the neuron k of the hidden layer and $w_k$ is the weight of the connection between the neuron k of the hidden layer and the final output, that is, the computed $A_T$. Finally, the absolute value of $C_i$ was expressed as a percentage of the sum of all $C_i$.

In addition to the test in the GLODAPv2 independent set, the network potential was tested on five ocean time-series in different oceanographic regimes that were not included in GLODAPv2: Hawaii Ocean Time-Series (HOT), Bermuda Atlantic Time-Series Study (BATS), European Station for Time-Series in the Ocean at the Canary Islands (ESTOC), Kyodo North Pacific Ocean Time-Series (KNOT) and K2.

GLODAPv2 contains quality controlled measurements in all ocean basins from the 1970s until 2013 (Olsen et al., 2016). However, winter data are scarce to absent in some high latitude regions because adverse weather conditions prevents field activities in that season (Fig. 3). In surface ocean, this temporal bias can be avoided with the help of the subsurface data from seasons with sufficient samples. Vázquez-Rodríguez et al. (2012) demonstrated how the subsurface ocean layer in the Atlantic Ocean can retain the footprint of the water mass formation from the preceding winter in the following months and, therefore, of the surface conditions. The winter relationship between inputs and $A_T$ needed to produce an all-season surface climatology are mostly preserved in this subsurface layer. The validity of this hypothesis was tested in other regions (Fig. 3) following Vázquez-Rodríguez et al. (2012). These areas were chosen based on the non-availability of $A_T$ data in two or more consecutive months in the same oceanographic regime as the colored area in Fig. 3.

To reinforce the previous test and to assess the ability of the neural network in overcoming the lack of winter data in other depths, a neural network was trained excluding all winter data in GLODAPv2 (GLODAPv2_nowinter) and tested in the excluded and independent winter dataset (GLODAPv2_winter). The procedure to create and to train the network was the same as described previously.

### 2.4 Climatology

Finally, we generated a 1ºx1º global monthly climatology of $A_T$ on 102 depth levels from the objectively analyzed climatological fields of WOA13 (Locarini et al., 2013; Zweng et al., 2013; Garcia et al., 2014a; Garcia et al., 2014b). From this database, the same input variables as in the training stage were selected to estimate $A_T$ from the relationships learned by the network. This final product was compared with the monthly sea surface climatologies of $A_T$ of Lee et al. (2006) and Takahashi et al. (2014). Furthermore, the annual mean was compared with the annual mapped climatology by Lauvset et al. (2016). The availability in Lauvset et al. (2016) of the climatologies of the variables used as inputs in the network were used to test how the network represents their climatology of $A_T$ and to evaluate the sources of the possible differences.

### 3 Results and discussion

### 3.1 Neural network analysis

The lowest RMSE was reached in the training and in the test sets when 128 neurons were used (Fig. S1). Similar RMSE values for both sets (training: 8 µmol kg$^{-1}$ vs test: 8.5 µmol kg$^{-1}$; Fig. S1 and Fig. S2) showed that no overfitting occurred, and that

the network generalizes well. The two training techniques did not show significant differences (Table 1). The Levenberg-Marquardt algorithm was selected for its higher computing speed. We also found no improvement by increasing the number of data points in the dynamic training set. The main reason is perhaps the random division of the datasets. All possible relations the network can learn could be represented using only 70% of the static training set, that is, 63% of the GLODAPv2. This result suggests the necessity to include other input variables rather than more data to improve the network mapping.

Samples with residuals beyond ±3RMSE are 1% of the GLODAPv2 dataset. The spatial distribution of these samples (Fig. S3) show that they are confined to certain areas, mainly in the ocean surface (Fig. 4). Most are in the Northern Hemisphere (Fig. S3 and Fig. 4). Specifically, 64% are from latitudes north of 60ºN (Table S1). In this area, 6.5% of GLODAPv2 samples have residuals beyond ±3RMSE and 83.1% of these samples are from the upper 100m (Table S2). In these depth and latitude ranges, the samples with high residuals make up 14% of the GLODAPv2 samples here and they typically have salinities lower than 34 (Table S3; Fig. S3). A monthly analysis in the previously indicated ranges shows that the largest number of samples with residuals beyond ±3RMSE are from the summer months. About 15-19% of all the samples from this season in this area have residuals higher than ±3RMSE (Table S4).

The previous results show that the Arctic Ocean is the region with the largest RMSE, although the network computes well most of the measured $A_T$ in this area. However, the low availability of winter data, the ice-sea dynamics and the transport of $A_T$ by the rivers (Fig. S4) could alter the presence of the surface winter conditions in the summer subsurface layer shown by Vázquez-Rodríguez et al. (2012) in other areas and generate a temporal bias in the climatology. The high discharge of high $A_T$ waters by the rivers in the summer (Cooper et al., 2008; Shiklomanov et al., 2018; Fig. S5) generates the greatest errors and shows how the network fails to model riverine $A_T$.

In further detail, many of the samples with residuals beyond ±3RMSE are located in the Beaufort Sea (66°N - 80°N, 140°W-180°W). Here, Takahashi et al. (2014) also found the largest RMSE (60.5 µmol kg$^{-1}$; 57.6 µmol kg$^{-1}$ applying their regression on GLODAPv2) of their SSS-PALK relations in the upper 50m of the water column. This area is specifically complex for the model surface $A_T$ because of significant river runoff having high and possibly variable $A_T$ concentrations (Fig. S4 and S5; Anderson et al. 2004; Cooper et al. 2008). Therefore, in spite of the good reproduction of $A_T$ for the most samples, one should be cautious with the results in this zone and for the entire Arctic Ocean.

The North Sea also contains many samples with large residuals. Those samples shallower than 100m and close to the coasts surrounding this sea do not have an accurately computed $A_T$ (Fig. S3 and Fig. S4). Some studies have shown the complexity of the processes occurring in this shallow sea where the high river runoff also has elevated levels of $A_T$ (Fig. S4; e.g., Hoppema, 1990; Artioli et al. 2012). Hence, the same caveats as for the Arctic Ocean should be made.

In general, the network mainly fails to compute $A_T$ in some samples of areas with rivers carrying significant amounts of $A_T$ to the ocean. The samples beyond ±3RMSE represent 23% and 9.4% of the total above 100m for the Beaufort Sea and the North Sea respectively. The inclusion of predictors related to riverine $A_T$ (and probably to ice melt) could improve the computation in these areas. Although one should be cautious, these zones still should be taken into account and be represented in the climatology since most of the samples have a well-computed $A_T$.

In the global ocean surface layer, the RMSE obtained with the neural network approach is lower than that obtained by previous studies on generation of monthly climatologies (Table 2 and 3). In the past, relationships between SST and SSS with $A_T$ by Lee et al. (2006) have been shown to produce the lowest RMSE (area-weighted RMSE of 8.1 µmol kg$^{-1}$) in the $A_T$ computation to create a monthly climatology. However, applying the relations of that study to GLODAPv2, the obtained weighted RMSE is higher than the one from the neural network (Table 2). Neural network approach obtained a better fit in all the areas defined in the study of Lee et al. (2006) (Table 2). NN±3RMSE improves the results obtained with the NNGv2 in almost all the regions, being the most remarkable the Equatorial Upwelling Pacific. However, the difference in the weighted RMSE of the two networks is not significant.

Similar to the previous case, the analysis of the error in the areas defined in Takahashi et al. (2014) also shows a better fit of the neural network (Table 3). Except for the zone with the lowest number of samples (Red Sea), the other 26 areas have a lower RMSE when the $A_T$ is computed by a neural network. The NN±3RMSE improves the fitting of the NNGv2 in the non-Arctic areas. The $A_T$ computed in the zones defined in the Arctic have higher RMSEs in the two approaches (Takahashi et al. (2014) and this study; Table 3). As discussed before, the Beaufort Sea is the zone with the highest RMSE. The inclusion of this area in calculating a global RMSE raises its value considerably. The NN±3RMSE has a higher global weighted RMSE because of the exclusion of most of the samples in this area to train this network. However, the weighted RMSE calculated excluding this area shows again a non-significant difference between the two networks (Table 3).

The results of the two networks clearly show how this fitting technique computes $A_T$ more accurately than the other methods used in studies on the generation of monthly climatologies. The non-linear nature of the neural networks used in this study and the inclusion of multiple predictor variables related to the $A_T$ variability are the main reasons for a good fit. Furthermore, we only used one neural network for the entire ocean. This has the advantage of obtaining the computed $A_T$ anywhere in the ocean in only one step. No "patches" or smoothing are needed between different zones in the climatology as there are in previous studies. Finally, the NNGv2 has been chosen to generate the climatology. Although NN±3RMSE computes $A_T$ with lower errors than NNGv2 in the non-Arctic areas, in a global view the improvement is relatively small (Weighted RMSE in Table 2 and Table 3). In order to include the Arctic in the climatology, the better fit in this area with the NNGv2 approach makes it the best candidate. In any case, the NN±3RMSE is also offered to the users who want to obtain a climatology or $A_T$ computations in a specific area where this network computes $A_T$ better than NNGv2 (e.g., Equatorial Upwelling Pacific, Southern Ocean, etc.).

The newest methods in the $A_T$ computation (LIARv2: Carter et al., 2018; CANYON-B: Bittig et al., 2018) model the GLODAPv2 $A_T$ with higher errors than the NNGv2 (Table 4). An analysis in a GLODAPv2 subset excluding the samples where the 2nd (Olsen et al., 2016) QC was not done for $A_T$ shows a reduction of the error in these three methods, being CANYON and NNGv2 the lowest (Table 4). All the equations are used to compute $A_T$ in the GLODAPv2 dataset when the computation is allowed to be made by the equation with the lowest uncertainty in each sample (Carter et al., 2018). The most used equations are 10 (S, N, Si), 15 (S, AOU) and 14 (S, N), which are used in about 50% of the samples. The equation that used all the input variables (1) is only used to model 3% of the GLODAPv2 samples. Surprisingly, when only this equation is used to compute $A_T$ in GLODAPv2 dataset, the error is lower than those obtained with the free election of the equation based on the lowest uncertainty. That result shows the potential of include all possible inputs related with the $A_T$ variability, although reasonable results can also be reached with the equations that do not use all the input variables. CANYON-B is an example of using relatively few input variables (position, time, temperature, salinity and oxygen) and getting good results (Table 4). Probably, the non-linear character of the neural networks, like the one used in CANYON-B, gives the high potential to this kind of methods to fit complex functions even with few input variables. However, the NNGv2 designed in the present study is the best option to model more GLODAPv2 data better than the other methods (lower RMSE) and therefore to use the mapped inputs-output relation in order to create the monthly climatology. The availability of all the variables used as inputs of the NNGv2 in WOA13 also contributes to make this method the best choice. Furthermore, methods like CANYON-B which include a predictor that explicitly accounts for the time variation of $A_T$ (decimal year in the case of CANYON-B), are not suitable to build a monthly climatology since they generate an unrealistic seasonal amplitude, at least at high depths. This has been checked used WOA13 monthly climatologies (temperature, salinity and dissolved oxygen) as inputs of CANYON-B to compute $A_T$ at different depth layers. As an example, in the 3000m depth layer, seasonal amplitudes up to 40 µmol kg$^{-1}$ were obtained in large areas mainly located between 30 and 60ºS.

The NNGv2 seems to associate the $A_T$ variability to the predictor variables in coherence with the processes that contribute to it. The relative importance of these variables depicted in Fig. 5 shows that salinity is the most influential variable, followed by dissolved oxygen and nutrients. In the surface layer, where $A_T$ variability is the largest, different studies showed how changes in salinity are highly correlated with this variability (Millero et al., 1998; Takahashi et al., 2014). The organic matter cycle also has a significant component in the $A_T$ variability (Kim and Lee, 2009). The formation and degradation of organic matter is reflected through both oxygen and nutrients variations. The network seems to capture the $A_T$ variability because of the organic matter cycle giving a second place in importance to these variables. The third group of variables in the ranking of importance is comprised by depth and temperature. The former variable could be associated to the $A_T$ variability accounting for the variation produced by the $CaCO_3$ cycle and the processes acting through the global ocean circulation. The latter has also been associated to the $A_T$ variability as a proxy of both the $CaCO_3$ and the organic matter cycles (Lee et al., 2006). Finally, the minor contribution of the variables of horizontal sampling position could help to separate the different relations shown by previous studies in different ocean areas (Lee et al., 2006; Takahashi et al., 2014).

**3.2 Time-series validation**

The network can compute $A_T$ well at 5 different ocean time-series stations. Low RMSEs and high coefficients of determination
($r^2$) were obtained (Table 5). The bias is relatively low in the three time-series with the highest number of data (HOT, BATS and ESTOC). The $A_T$ computed by the NNGv2 at KNOT and K2 is slightly higher than the measured one, probably because of the influence in the $A_T$ variability of some variable not included as an input of the network (although an offset in the measurements of any of the inputs could also give this result). Summed to the previous test, the statistics obtained in this independent test with a good seasonal time resolution shows the good generalization of the NNGv2.

The ability of NNGv2 to capture surface $A_T$ variability is exemplified in Fig. 6. The other largest time-series also show a good agreement between the computed and the measured seasonal $A_T$ in this surface layer (RMSE HOT: 5.3 µmol kg$^{-1}$; RMSE ESTOC: 4.2 µmol kg$^{-1}$). In general, $A_T$ measured in each month of the year are well modeled by NNGv2 (inner charts in Fig. 6). The same holds for other depth layers (Fig. 7, panels in left column). Only some extreme values are not fully captured but almost all the trends between months are well represented. The differences may be caused by bias in measured $A_T$ or some of
the input variables; they may also be due to an under/overestimation of the network. Furthermore, the time-series areas are not fully represented in all months in GLODAPv2 so that NNGv2 might not represent seasonality well. However, the network computes $A_T$ in any month with a very low error. This shows again the potential of the generalization of a well-designed neural network.

The NNGv2 also has the capacity to increase the number of $A_T$ data in the time-series. In many samples, $A_T$ was not measured
but the other input variables needed for the NNGv2 are available. Therefore, the computed $A_T$ has a higher temporal and spatial resolution than observations only. This enables the computation of more reliable trends than with the less frequently measured $A_T$ and allows the identification of possible high frequency changes. The improvement in resolution is especially visible in the longer time-series: HOT and BATS (Fig. 7). In the former we increased the number of $A_T$ data from 3852 to 14089 and in the latter from 3033 to 11342 (Fig. 7, panels in central column).

The LIARv2 and CANYON-B methods to compute $A_T$ also model the time-series data quite well (Table 6). Significant differences among the three methods are obtained in HOT and ESTOC. In HOT, NNGv2 and CANYON-B reach a better fit of $A_T$ than LIARv2 suggesting that a non-linear technique is more adequately to model $A_T$ in this area (Table 6). In ESTOC, NNGv2 and LIARv2 are the best options to model the $A_T$ variability (Table 6). Here, the $A_T$ computed with LIARv2 with the option of the free equation choice activated results in a greater election of the equations that include nutrients as predictors.
This result show how in this area the inclusion of nutrients as predictors contributes to improve the model of $A_T$. Like NNGv2, both methods have a considerable bias in K2 and KNOT (Table 6) that reinforce the two reasons suggested previously.

**3.3 Subsurface Layer Hypothesis**

We found that the optimal depth range of the subsurface layer defined by Vázquez-Rodríguez et al. (2012) for the North Atlantic Ocean (100-200 m) must be modified in other regions. In the area analyzed in the Indian Ocean (Fig. 3), the subsurface layer hypothesis is verified in the same depth range of that study. However, the other areas (Fig. 3) show that the range of the subsurface layer is in the range of 50-100 m. The different strengths of deep mixing and convection in winter could explain this fact.

The properties analyzed in the four areas defined in Fig. 3 show, as expected, a higher monthly variability in the ocean surface than in the subsurface layers. The seasonal variability depicted in Fig. 8 will likely be typical of a larger region within a similar oceanographic regime for each defined area. The surface winter conditions of the analyzed properties are quite similar to those in the subsurface layer during, at least, one of the four consecutive months following winter in all areas (Fig. 8).

The optimal number of neurons in the network trained with GLODAPv2_nowinter dataset to reinforce the subsurface layer hypothesis and to assess the layers below surface ocean was 100. The reduction of the number of neurons compared to the previous networks was because this new dataset contains less data. Thus, maintaining or increasing the number of neurons would produce overfitting. This new network provides statistics in the GLODAPv2_nowinter dataset similar to those of the network used to create the climatology (NNGv2) in GLODAPv2 dataset (Table 1 vs Table 7). But, of greater importance are the statistics resulted from the GLODAPv2_winter dataset (Table 7) which reinforce the subsurface layer hypothesis. The low error reached in this independent winter dataset shows how the network is able to obtain the winter relations in any depth from the function fitted with data from other seasons. Therefore, the lack of winter data in different regions does not automatically mean that the climatology will be biased towards the more sampled seasons.

### 3.4 Climatology

The monthly climatology of $A_T$ is based on the relations obtained in the training procedure of the neural network applied to the WOA13 monthly climatological fields. We have demonstrated that the $A_T$ computed by the two offered neural networks agrees reasonable with the measured $A_T$ when the inputs associated to it are passed through the networks, i.e. the relations obtained from GLODAPv2 in the training stage are robust. Therefore, the $A_T$ patterns in the climatology are forced by the patterns of the WOA13 variables used as inputs. The monthly climatology can be found in a netCDF file at the data repository of the Spanish National Research Council (CSIC; doi: http://dx.doi.org/10.20350/digitalCSIC/8564) together with a video of the monthly variation at the surface and in three longitudinal sections of the three main oceans.

The distribution of the surface annual mean $A_T$ (Fig. 9) is similar to that shown in previous climatologies (e.g., Lee et al. 2006; Takahashi et al. 2014; Lauvset et al. 2016). Not surprisingly, there is a high correlation with the salinity distribution and, consequently, with the evaporation-precipitation patterns. The largest values in the surface layer occur in the Mediterranean Sea, Red Sea, and in the subtropical gyres of the Atlantic and South Pacific Oceans, all of them prevailing throughout the year

in the monthly climatology. At depth, these maxima are all present at least up to 150m (Fig. 9). Below 700m, the Pacific and Indian Oceans show higher $A_T$ concentrations than the younger waters of the Atlantic (Fig. 9). Furthermore, features such as

the high-$A_T$ Mediterranean Water entering the Atlantic Ocean are captured in the climatology (Fig. 9, 1000m chart, black circle). In general, the patterns agree with the main ocean processes responsible for the $A_T$ variability as explained previously.

The seasonal amplitude of sea surface $A_T$ (Fig. 10) is generally in agreement with that obtained by Lee et al. (2006). The highest amplitudes are in the north equatorial zone, in the Arctic Ocean and in coastal zones, i.e., at locations where there are rivers with a large water discharge (like the Amazonas, Congo, La Plata or Arctic rivers). The seasonal amplitude of the surface

salinity (Fig. S6) can explain most of the variability in the seasonal amplitude of $A_T$. In areas with a large seasonal amplitude of salinity (more than 1 unit; mainly the Arctic Ocean and coastal zones near rivers with high discharge), this variable linearly explains 76% of the seasonal amplitude $A_T$ variability. However, the seasonal amplitude in the Arctic Ocean should be taken with caution due to the difficulty to accurately model this complex zone, as discussed previously. Despite the presence of high levels of $A_T$ in some river mouths in the melting months, the $A_T$ carried by the rivers could be not represented in the climatology

and this can enhance the seasonal cycle due to an underestimated value in low salinity waters with high riverine $A_T$. On the other hand, in areas with a low seasonal amplitude of salinity (less than 1 unit; mainly oceanic areas and coastal regions without rivers with high discharge) about 61% of variability is linearly explained. This result shows the importance of the inclusion of other predictors besides salinity in the network and the non-linearity of the method proposed in this study to explain nearly all the $A_T$ variability.

The seasonal amplitude of $A_T$ is progressively reduced at depth (Fig. S7). The changes in the variables which influence the changes in $A_T$ are smaller than in the surface layer or null causing this reduction. The seasonality disappears almost completely below 500m depth; not surprising due to the lack of seasonal resolution in the climatologies of nutrients in WOA13 below this level. Some patches of variability are present likely because of a conjunction of the error of the network and the monthly changes in the other WOA13 input variables. In addition, they could also come from the learning stage since the training data

present monthly variations of up to ~10 µmol kg$^{-1}$ for the same area, even at depths greater than 1000m.

Although it was shown that the neural network can accurately compute $A_T$ in both GLODAPv2 and time-series datasets, the quality of WOA13 data also determines the robustness of the climatology is. Unfortunately, WOA13 does not offer uncertainty fields associated to the objectively analyzed climatologies to compute a coherent estimation of the uncertainty in the $A_T$ climatology. Therefore, the climatological values offered in this study should be evaluated by comparing them with

observations in a monthly average over many years. This can only be done at the locations of time-series with representative amounts of data; Fig. 11 shows this analysis at surface. At both the BATS and HOT time-series, the differences between the averaged measured $A_T$ (Fig. 11, red line) and the climatology (Fig. 11, yellow line) are quite low. The comparisons are better when $A_T$ is computed by NNGv2 using as inputs the measured values in the time-series (Fig. 11, purple line). The differences of the two comparisons show the differences in the input variables (WOA13 climatological fields vs time-series input data).

The previous results hold true also for other depth layers. A comparison of monthly profiles up to about 500m between the $A_T$ climatology obtained from WOA13 and the one from the averaging of the time-series data shows low differences. In BATS, the RMSE of this comparison ranges between 1.1 and 2.8 µmol kg$^{-1}$ (mean RMSE of 2 µmol kg$^{-1}$) and the bias between 0 and 4.7 µmol kg$^{-1}$ for all months. In HOT, the RMSE of this comparison ranges between 5 and 10.5 µmol kg$^{-1}$ (mean RMSE of 6.4 µmol kg$^{-1}$) and the bias between -0.3 and 6.3 µmol kg$^{-1}$ for all months. The climatological measured data are for the periods

between 1991 and 2015 (BATS) and 1989 and 2016 (HOT) and WOA13 data are supposed to cover a larger range. Despite this time difference, the $A_T$ climatology represents quite accurately the measured values averaged in each month.

    Compared to the other climatologies, the surface annual mean $A_T$ of this study is closer to that of Lee et al. (2006) (Table 8). This is likely because temperature and salinity are included as non-linear predictors of $A_T$. In Takahashi et al. (2014), $A_T$ derives from the linear regression between PALK and one predictor (salinity) and in the Lauvset et al. (2016) study, DIVA

(Data-Interpolating Variational Analysis; Troupin et al., 2010) was used. Furthermore, the transfer of our climatology to the coarser grid of Takahashi et al. (2014) for the comparisons may enhance dissimilarities.

    The comparison of the monthly values of our climatology and the other climatologies available at the same time frequency (Table 9) shows the greatest similarity of ours and that of Lee et al. (2006). The reasons given above may also hold here. In addition, part of the differences between the comparisons may originate from the different versions of the WOA used in each

study (Lee et al., 2006: temperature and salinity from WOA01; Takahashi et al., 2014: salinity from WOA09 and nitrate from WOA94; this study: all inputs from WOA13).

    In general, the surface spatial patterns of the differences between the annual mean of our $A_T$ climatology and the three other ones under consideration are not correlated (Figure S8). Compared to Takahashi et al. (2014), the largest differences are in the Beaufort Sea and in three zonal bands: 54-60º S, 8-28º N and 40-60º N (Fig. S8a). The Pacific Ocean has the highest

dissimilarities in these three bands. In general, the Atlantic Ocean and the Indian Ocean have the smallest differences. The largest differences in these two ocean basins are mainly located close to the river mouths. It shows how the different parametrizations of the $A_T$ diverge highly at low salinities. On the other hand, the major differences with Lee et al. (2006) (Fig. S8b) are surrounding North America's Pacific coast, the area of influence of the Amazon river, the zone between both the Niger and the Congo rivers and the North Sea. In the open ocean there are some wide areas where the differences are

remarkably high. They are mainly in the South Pacific. It should also be noted that the transition zone between the 1 ((sub)tropics) and 2 (equatorial upwelling Pacific) areas defined in the study of Lee et al. (2006) generates a discontinuity in the difference map. Finally, the largest differences with Lauvset et al. (2016) (Fig. S8c) are less localized. The Arctic Ocean and the Pacific sector of the Southern Ocean are the areas where there is a large spatial continuity in the differences.

    An important cause of the differences between the climatologies stems from the use of different inputs to generate them. As

an example, this can be seen when the climatologies of Lauvset et al. (2016) are used as input variables to compute $A_T$ with

the neural network instead of the WOA13 data (Fig. 12). In the surface layer, a considerable reduction of the RMSE (15.7 to 12.3 µmol kg$^{-1}$) and an increase of the r$^2$ from 0.91 to 0.95 are obtained (Fig. 12). In the deeper layers, the differences are progressively decreasing. The values of the RMSE of the comparisons like those in Fig. 12 but below 250m are in the range of 4 to 6 µmol kg$^{-1}$ and the improvement caused by the inputs usage is reduced to around 1 µmol kg$^{-1}$. This last result shows an increasing similarity between WOA13 climatologies and Lauvset et al. (2016) climatologies with increasing depth. However, and to be consistent, it is recommended to use the A$_T$ climatology corresponding with the other inputs used in the studies that arise from these products.

**4 Data availability**

The climatology and the two neural networks designed in this study are available at the data repository of the Spanish National Research Council (CSIC; doi: http://dx.doi.org/10.20350/digitalCSIC/8564).

**5 Conclusions**

A neural network to compute A$_T$ anywhere in the ocean has been presented. As evaluated by the RMSE between the measured and the computed data, the neural network approach presented in this study offers increased precision compared to most of the approaches in previous studies. Furthermore, the global relationship between A$_T$ and input variables was obtained from a higher number of quality-controlled data than before in the generation of a monthly climatology, with a greater temporal and spatial resolution. We have demonstrated how one single global algorithm is able to compute A$_T$ satisfactorily for the entire global ocean. This has enabled us to generate a monthly climatology without the need to use smoothing techniques between different oceanic areas. Furthermore, the seasonal variability in depth is more realistic than the one computed by other methods that overestimate it.

The validation using different independent datasets demonstrates the good network generalization. In addition, the spatiotemporal A$_T$ variability is well captured by the network as shown in time-series validation. Therefore, the obtained climatology using WOA13 inputs should reflect this variability due to the good network performance to new independent data.

We offer this global monthly climatology of A$_T$ to the scientific community for advancing the understanding of the ocean carbon cycle. Our new climatology may particularly be useful as input to modeling efforts. It is worthwhile mentioning that the networks offered here are also useful to obtain A$_T$ values for samples where the inputs for the neural network are present.

**6 Author contributions**

DB, FFP and AV designed the study. The manuscript was written by DB and revised and discussed by all the authors. The dataset of the climatology and the neural networks were created by DB.

## 7 Competing interests

The authors declare that they have no conflict of interest.

## 8 Acknowledgements

This research was supported by Ministerio de Educación, Cultura y Deporte (FPU grant FPU15/06026), Ministerio de Economía y Competitividad through the ARIOS (CTM2016-76146-C3-1-R) project co-funded by the Fondo Europeo de Desarrollo Regional 2014-2020 (FEDER) and EU Horizon 2020 through the AtlantOS project (grant agreement 633211). The authors want to thank the comments of Siv K. Lauvset to improve the manuscript.

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

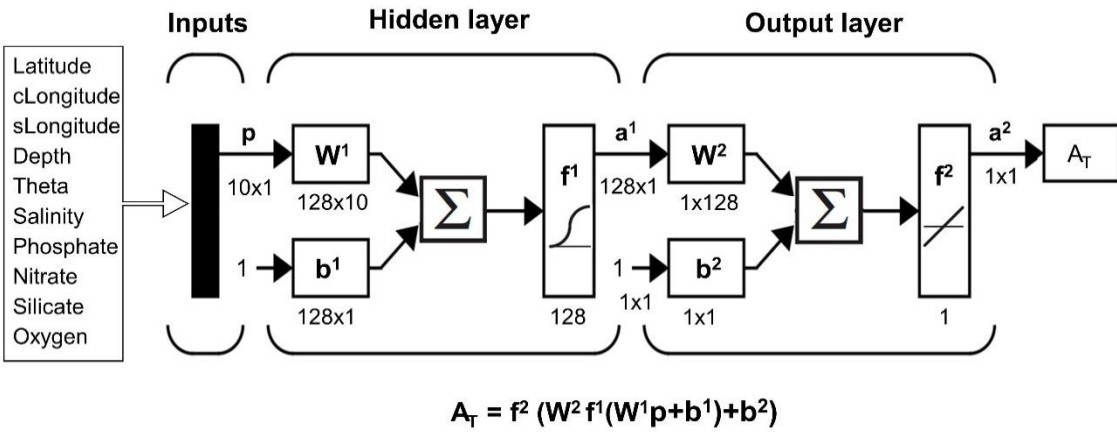

$$A_T = f^2 (W^2 f^1 (W^1 p + b^1) + b^2)$$

**Figure 1: Neural network configuration. The notation is in agreement with Hagan et al. (2014). Theta: potential temperature; p:**
**input vectors; W: weight matrix; b: bias matrix; $\sum$: sum; f: transfer function; a: output matrix. The superscripts indicate the number of the layer. The c and s preceding month and longitude variables represent cosine and sine (See equations below)). The dimensions of the matrices are for an individual sample. Modified from Hagan et al. (2014).**

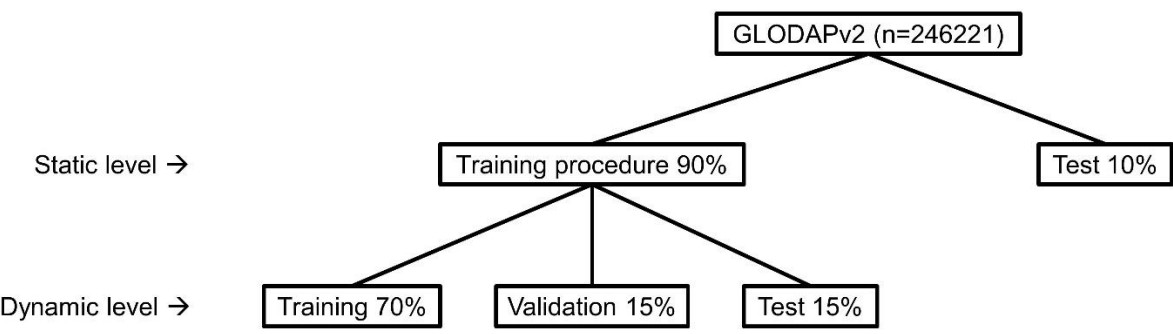

**Figure 2. Division of the data for the training of the network. The data in the sets of the static level is the same for all the networks**
**to train. The data in the sets of the dynamic level is randomly selected for each network to train.**

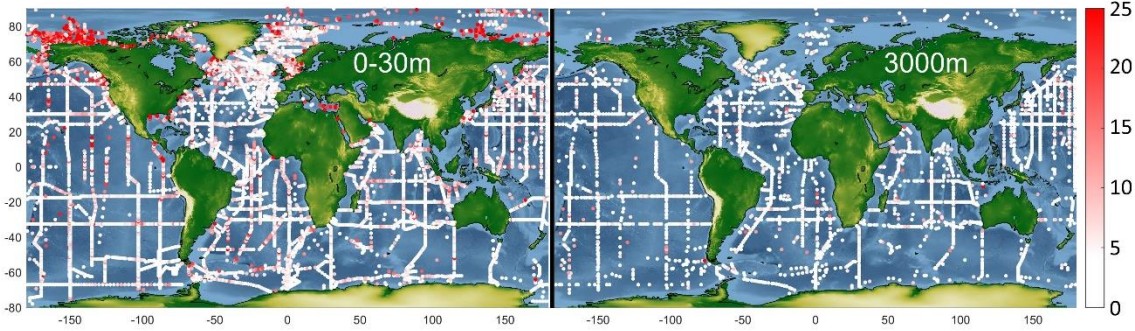

**Figure 3: Locations of GLODAPv2 data used in this study presented by month of observation (red dots). Areas where subsurface layer hypothesis was evaluated are shown as colored rectangles.**

**Figure 4: The absolute differences between GLODAPv2 $A_T$ and NNGv2 $A_T$. Left: samples in the layer 0-30m. Right: samples in the layer 2950-3050m.**

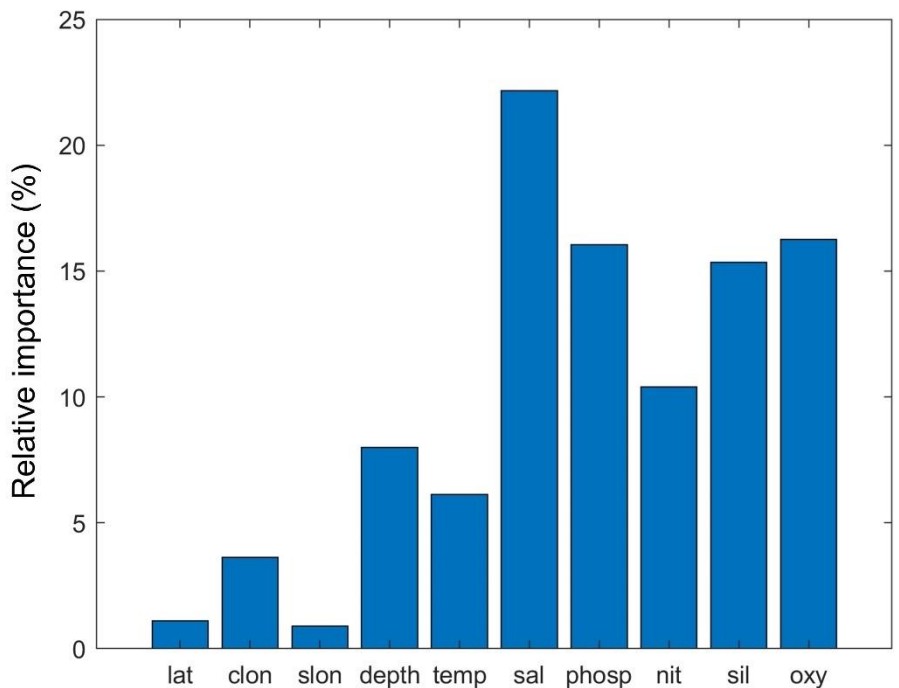

**Figure 5: The relative importance of the predictor variables for the NN. lat: latitude; clon: Eq. (3); slon: Eq. (4); temp: potential temperature; sal: salinity; phosp: phosphate; nit: nitrate; sil: silicate; oxy: oxygen.**

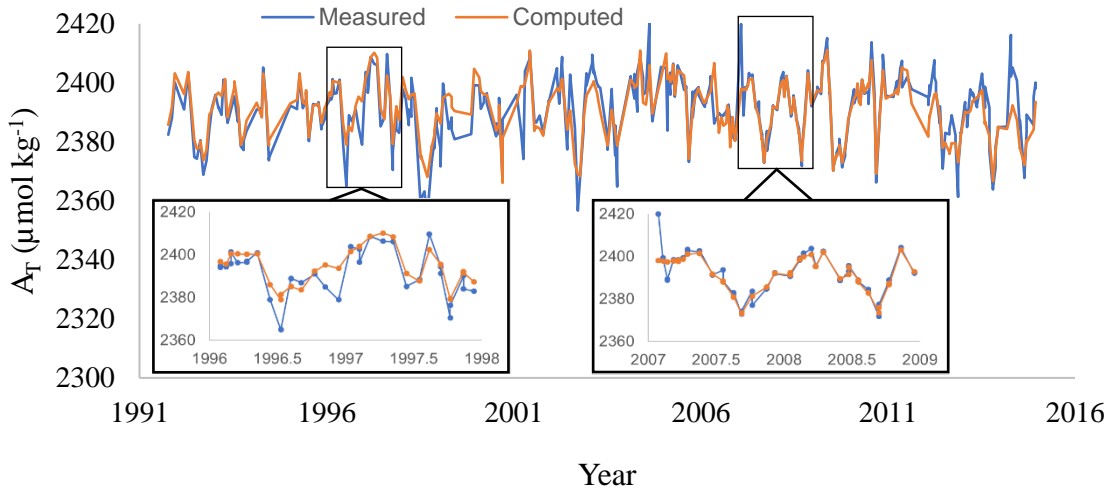


**Figure 6: Comparison of measured and computed $A_T$ for the depth range 0-10 m at time-series station BATS. The RMSE in that depth range for the whole time-period is 5.7 μmol kg$^{-1}$. The years 1996-1997 and 2007-2008 are amplified to show the monthly variations because they are the years with $A_T$ measurements in all the months.**

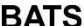

Figure 7: Left column: Computed $A_T$ for the upper 550m of the water column at the BATS and HOT time-series stations. Central column: Difference between measured and computed $A_T$. Colored dots show samples where $A_T$ was measured. Black dots show samples where $A_T$ was not measured but the network inputs were. Right column: Difference between measured and computed $A_T$ interpolated with Data-Interpolating Variational Analysis (DIVA; Troupin et al., 2010). This figure was made with Ocean Data View (Schlitzer, 2016).



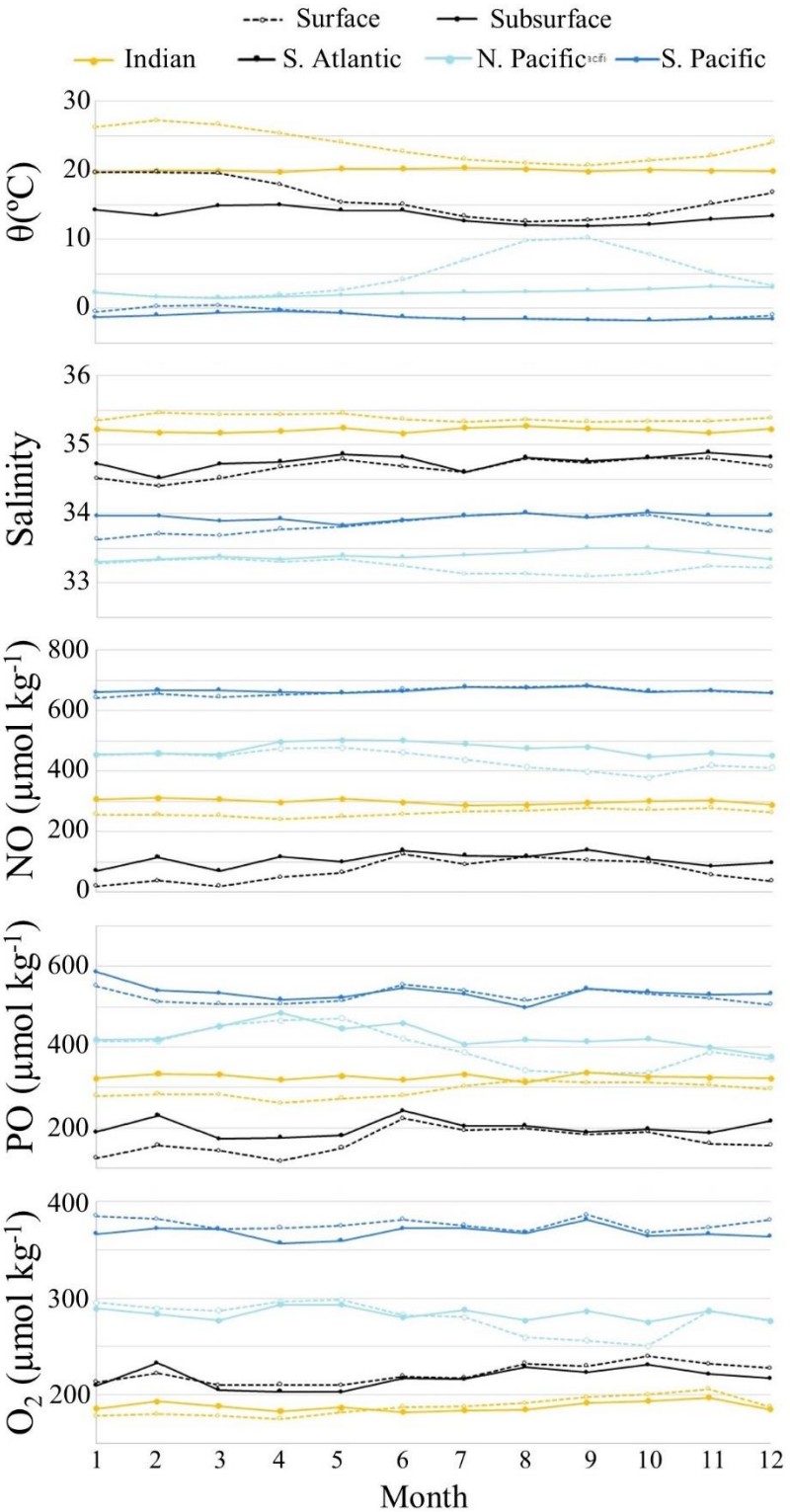

**Figure 8: Monthly variability of θ (potential temperature), salinity, NO = 9\*NO₃ + O₂ and PO = 135\*PO₄ + O₂ (defined according to Broecker, 1974) for different ocean basins. Data from WOA13 objectively analyzed monthly climatologies were averaged for each area defined in Figure 2. Each zone is displaced in each graph for a certain constant quantity of the variable for a better visualization, that is, the data shown are not the real values. Indian Ocean: 100-200m; South Atlantic, South Pacific and North Pacific: 50-100m.**

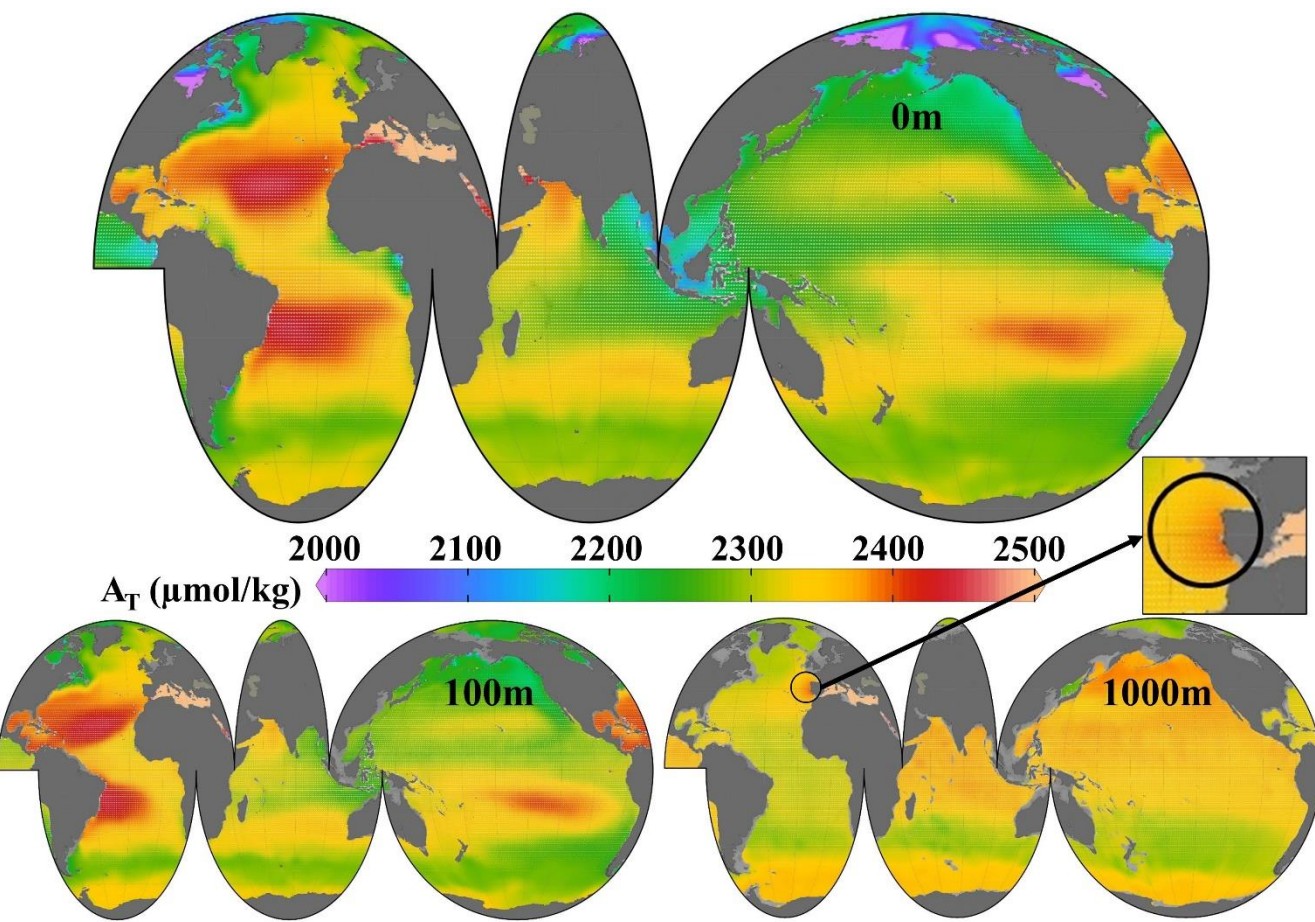

**Figure 9: Annual mean climatology of $A_T$ at 3 depths. Black circle in 1000m panel points out the area of influence of the Mediterranean Water in the Atlantic Ocean. This figure was made with Ocean Data View (Schlitzer, 2016).**

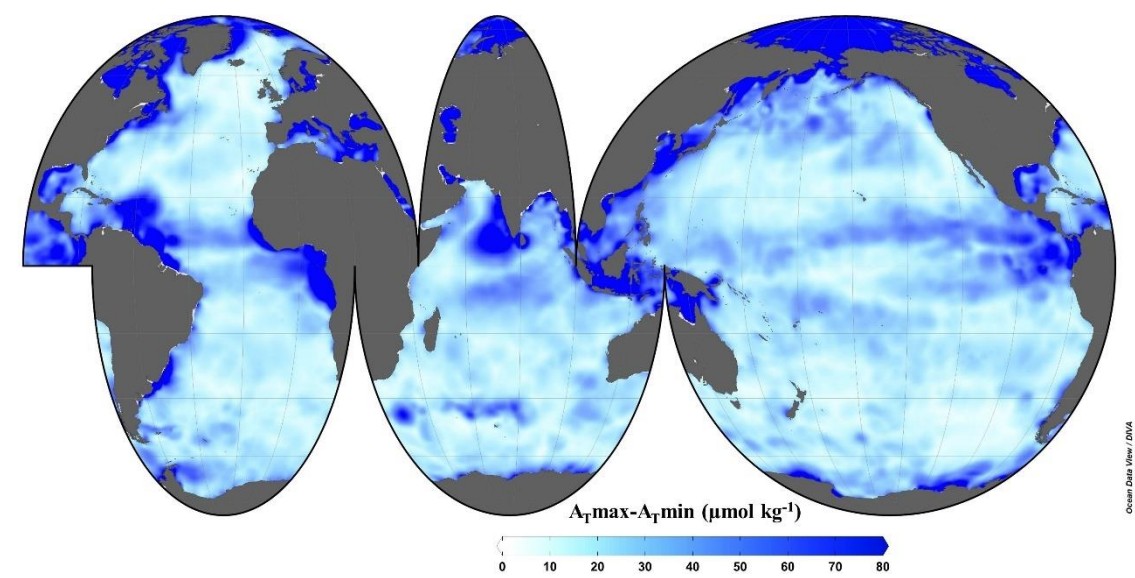

**Figure 10: Seasonal amplitude of sea surface A$_T$. This figure was made with Ocean Data View (Schlitzer, 2016).**

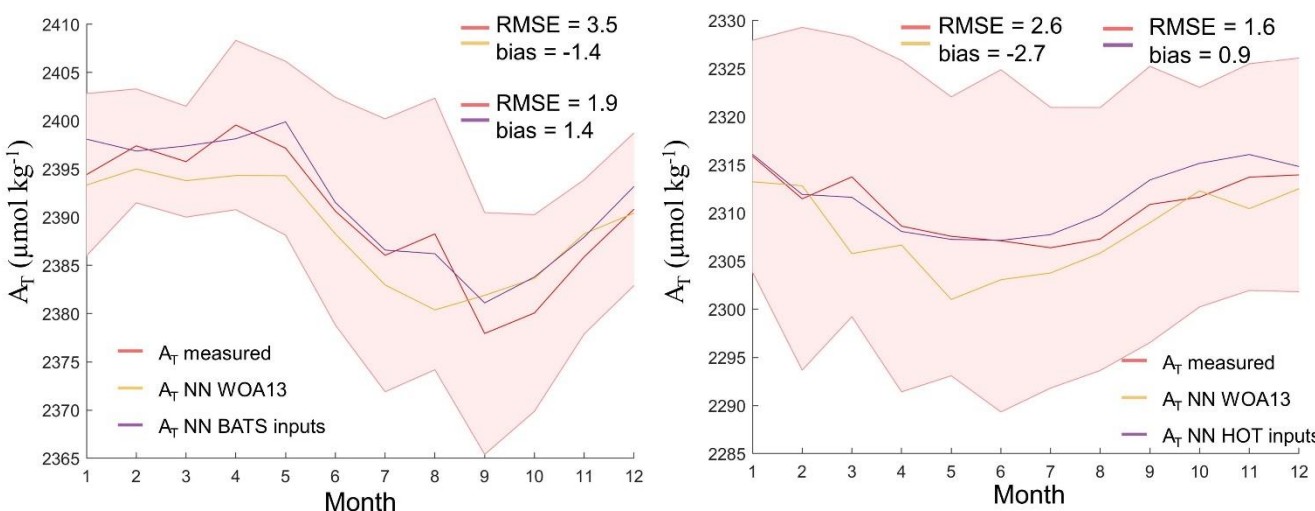

**Figure 11: Climatology of A$_T$ from measured data, from NNGv2 using measured data as inputs and from NNGv2 using WOA13 data as inputs at BATS (0-5 m; left panel) and HOT (0-30 m; right panel) time-series location. The shading represents the standard deviation of the average of the measured data. Units of RMSE and bias are µmol kg$^{-1}$**

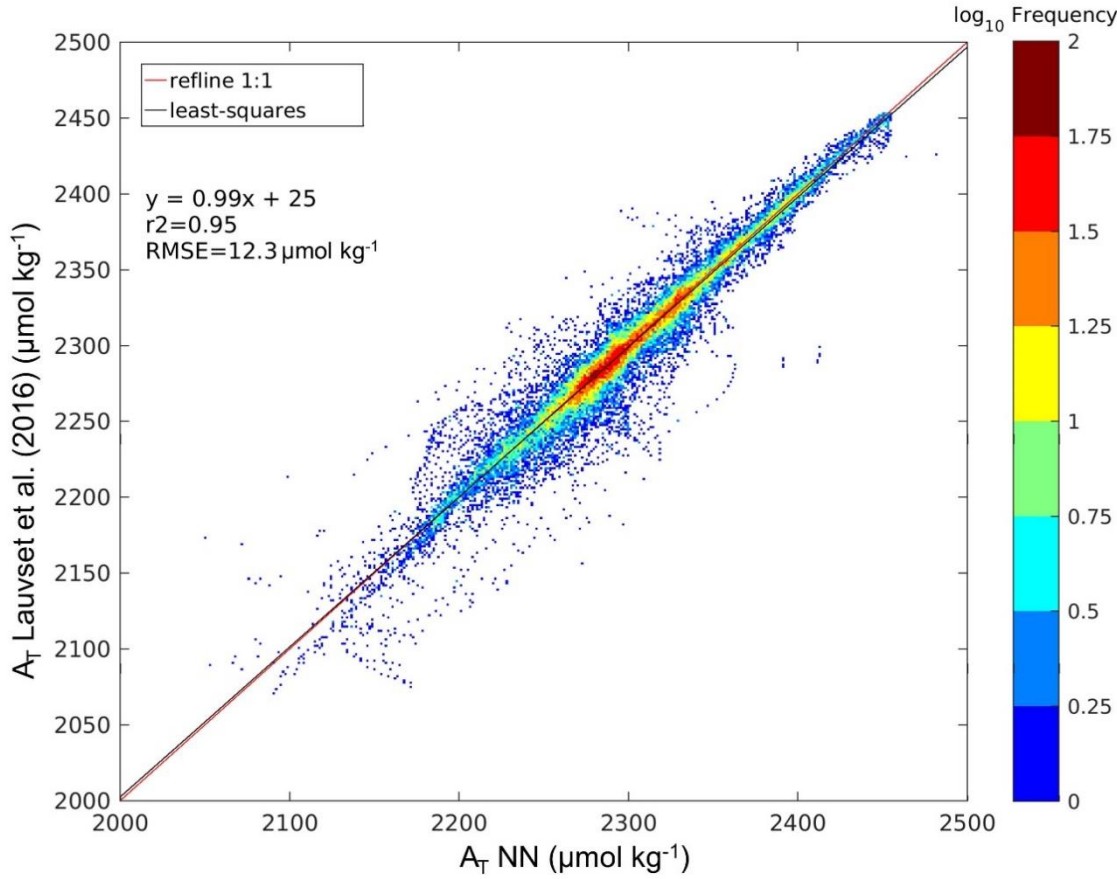

**Figure 12: Regression between A$_T$ computed with NNGv2 applied on the climatologies of Lauvset et at. (2016) and A$_T$ from Lauvset et at. (2016) at 0m. The graph is divided in pixels. The color of each pixel is determined by the number of points inside it. Note the logarithmic scale to account for the large amount of data.**

**Table 1: RMSE and bias between GLODAPv2 A$_T$ and A$_T$ computed by the neural network. n: number of samples. NNGv2: neural network trained with the initial dataset. NN±3RMSE: neural network trained with the dataset without samples with residuals beyond ±3RMSE. GLODAPv2: initial dataset. GLODAPv2±3RMSE: dataset without samples with residuals beyond ±3RMSE. lm: Levenberg-Marquardt. br: Bayesian Regularization**

| Approach | RMSE ($\mu$mol kg$^{-1}$) | bias ($\mu$mol kg$^{-1}$) | n |
|---|---|---|---|
| NNGv2_GLODAPv2 (lm) | 8.2 | 0.02 | 246221 |
| NNGv2_GLODAPv2 (br) | 8 | 0.03 | 246221 |
| NNGv2_GLODAPv2±3RMSE (lm) | 5.1 | -0.002 | 243754 |
| NN±3RMSE_GLODAPv2±3RMSE (lm) | 4.8 | -0.006 | 243754 |

Table 2: RMSE obtained by the relations of Lee et al. (2006), NNGv2 and NN±3RMSE over GLODAPv2. In bold the lowest RMSE in each area defined in Lee et al. (2006). To be consistent with the surface layer defined in Lee et al. (2006) the samples evaluated here are from above 20m (subtropics) and 30m (the rest).

| Areas defined in Lee et al. (2006) | RMSE | | | n |
|---|---|---|---|---|
| | Lee et al. (2006) | NNGv2 | NN±3RMSE | |
| North Atlantic | 13.5 | 12.3 | **12.1** | 2765 |
| North Pacific | 16.8 | 10.5 | **9.6** | 2087 |
| Equatorial Upwelling Pacific | 7.8 | 9.5 | **5.7** | 481 |
| Subtropics | 20.8 | **15.1** | 15.2 | 4309 |
| Southern Ocean | 10.1 | 5.9 | **5.3** | 3610 |
| Weighted RMSE | 15.3 | 11.1 | **10.6** | 13252 |

Table 3: RMSE obtained by the relations of Takahashi et al. (2014), NNGv2 and NN±3RMSE over GLODAPv2. In bold, the lowest RMSE in each area defined in Takahashi et al. (2014). To be consistent with the surface layer defined in Takahashi et al. (2014) the samples evaluated here are from above 50m.

| Areas defined in Takahashi et al. (2014) | RMSE ($\mu$mol kg-1) | | | n |
|---|---|---|---|---|
| | Takahashi et al. (2014) | NNGv2 | NN±3RMSE | |
| West GIN Seas | 29.2 | **10.4** | 11.8 | 623 |
| East GIN Seas | 11.6 | 9.5 | **9.0** | 990 |
| High Arctic | 24.9 | **15.2** | 16.2 | 594 |
| Beaufort Sea | 57.6 | **46.9** | 79.3 | 2086 |
| Labrador Sea | 27.7 | 22.7 | **22.1** | 736 |
| Subarctic Atlantic | 15.6 | **10.4** | 11.3 | 1041 |
| North Atlantic Drift | 7.7 | 7.9 | **7.2** | 1403 |
| Central Atlantic | 23.1 | **19.9** | 20.1 | 3276 |
| South Atlantic Transition Zone | 6.7 | 6.8 | **5.9** | 291 |
| Antarctic (Atlantic) | 7.5 | 5.8 | **5.2** | 727 |
| Kuroshio-Alaska Gyre | 16.2 | 10.8 | **9.8** | 1412 |
| North Central Pacific | 13.2 | 10.0 | **9.4** | 1224 |
| Okhotsk Sea | **5.4** | 7.8 | **5.4** | 20 |
| Central Tropical North Pacific | 9.3 | 7.3 | **7.0** | 1328 |
| Tropical East North Pacific | 30.9 | 11.2 | **10.3** | 308 |
| Panama Basin | 8.1 | 13.4 | **7.4** | 58 |
| Central South Pacific | 9.7 | 6.4 | **5.8** | 2834 |
| East Central South Pacific | 11.6 | 9.3 | **8.8** | 249 |
| Subpolar South Pacific | 8.2 | 5.2 | **4.6** | 431 |
| Antarctic (Pacific) | 4.9 | 4.3 | **3.0** | 524 |
| Main North Indian | 7.0 | 6.2 | **4.6** | 493 |
| Red Sea | **6.3** | 9.3 | 9.2 | 19 |

| | | | | |
|---|---|---|---|---|
| Bengal Basin | 8.9 | 7.8 | **6.3** | 96 |
| Main South Indian | 8.8 | 7.1 | **6.8** | 2536 |
| South Indian Transition | 7.9 | 5.4 | **3.8** | 330 |
| Antarctic (Indian) | 8.1 | 5.0 | **4.0** | 865 |
| Circumpolar Southern Ocean | 10.1 | 5.9 | **5.3** | 1970 |
| Weighted RMSE | 17.0 | **12.8** | 15.0 | 26464 |
| Weighted RMSE without Beaufort Sea | 13.5 | 9.9 | **9.5** | 24378 |

**Table 4. RMSE and bias obtained with NNGv2, LIARv2 (Carter et al., 2018) and CANYON-B (Bittig et al., 2018) in both GLODAPv2 dataset and GLODAPv2 dataset without the samples where AT QC was not done.**

| Approach | RMSE ($\mu$mol kg$^{-1}$) | bias ($\mu$mol kg$^{-1}$) | n |
|---|---|---|---|
| NNGv2_GLODAPv2 | 8.2 | 0.02 | 246221 |
| LIARv2_GLODAPv2 | 11.4 | 0.08 | 246221 |
| CANYON-B_GLODAPv2 | 10.2 | 0.1 | 246221 |
| NNGv2_GLODAPv2_onlyQC | 6.6 | 0.06 | 215332 |
| LIARv2_GLODAPv2_onlyQC | 8.2 | 0.06 | 215332 |
| CANYON-B_GLODAPv2_onlyQC | 6.8 | -0.04 | 215332 |


**Table 5: RMSE and bias between measured A$_T$ and neural network computed A$_T$. r$^2$ from the regression between measured A$_T$ vs computed A$_T$. The comparison was done for all the samples where the input variables and the A$_T$ were measured in the same water sample.**

| Time-Series | Location | RMSE ($\mu$mol kg$^{-1}$) | bias ($\mu$mol kg$^{-1}$) | r$^2$ | n |
|---|---|---|---|---|---|
| HOT | 22º45'N, 158º00'W | **5.8** | **-0.8** | **0.99** | 4010 |
| BATS | 31º40'N, 64º10'W | **6.2** | **-0.2** | **0.77** | 3033 |
| ESTOC | 29º10'N, 15º30'W | **3.3** | **0.6** | **0.99** | 1700 |
| KNOT | 44ºN, 155ºE | **4.7** | **-6.4** | **0.996** | 1234 |
| K2 | 47ºN, 160E | **3.1** | **-3.0** | **0.998** | 561 |

**Table 6: RMSE and bias between measured A$_T$ and the A$_T$ computed with both LIARv2 and CANYON-B methods. The comparison was done for the same samples evaluated in Table 5.**

| Time-Series | LIARv2 | | CANYON-B | |
|---|---|---|---|---|
| | RMSE | bias | RMSE | bias |

|  | (µmol kg$^{-1}$) | (µmol kg$^{-1}$) | (µmol kg$^{-1}$) | (µmol kg$^{-1}$) |
|---|---|---|---|---|
| HOT | **6.6** | **-0.6** | **5.8** | **-0.6** |
| BATS | **6.3** | **0.1** | **6** | **-0.4** |
| ESTOC | **3.4** | **0.8** | **4.2** | **3.2** |
| KNOT | **4.8** | **-6.6** | **4.5** | **-7.2** |
| K2 | **3** | **-3.0** | **3** | **-3.3** |

**Table 7: RMSE and bias obtained with the neural network trained without winter data in both GLODAPv2 dataset without winter data and GLODAPv2 dataset only containing winter data.**

| Dataset | RMSE (µmol kg$^{-1}$) | bias (µmol kg$^{-1}$) | n |
|---|---|---|---|
| GLODAPv2_nowinter | 8.7 | -0.3 | 225189 |
| GLODAPv2_winter | 6.8 | -0.4 | 21032 |


**Table 8: Comparison of four annual mean surface climatologies of A$_T$. *The Arctic Ocean and the Baltic Sea are not included in the comparisons for coherency reasons.**

| RMSE (µmol kg$^{-1}$)\r$^2$ | NNGv2 | Lauvset et al. 2016* | Takahashi et al. 2014 | Lee et al. 2006 |
|---|---|---|---|---|
| NN |  | 0.91 | 0.92 | 0.97 |
| Lauvset et al. 2016* | 15.7 |  | 0.90 | 0.92 |
| Takahashi et al. 2014 | 15.3 | 17.8 |  | 0.93 |
| Lee et al. 2006 | 8.0 | 14.6 | 12.4 |  |

**Table 9: Comparison between the three monthly climatologies of A$_T$.**

| Month | Lee et al. (2006) vs NNGv2 | | Takahashi et al. (2014) vs NNGv2 | | Lee et al. (2006) vs Takahashi et al. (2014) | |
|---|---|---|---|---|---|---|
|  | RMSE (µmol kg$^{-1}$) | r2 | RMSE (µmol kg$^{-1}$) | r2 | RMSE (µmol kg$^{-1}$) | r2 |
| January | 12.6 | 0.93 | 18.5 | 0.89 | 14.2 | 0.92 |
| February | 12.2 | 0.94 | 24.2 | 0.82 | 14.7 | 0.91 |
| March | 12.1 | 0.94 | 19.5 | 0.87 | 14.3 | 0.91 |
| April | 12.1 | 0.94 | 18.4 | 0.88 | 15.0 | 0.91 |
| May | 12.4 | 0.93 | 19.0 | 0.86 | 13.8 | 0.92 |
| June | 12.7 | 0.93 | 17.7 | 0.89 | 14.3 | 0.91 |
| July | 12.3 | 0.93 | 24.9 | 0.84 | 14.8 | 0.91 |
| August | 12.9 | 0.93 | 19.5 | 0.89 | 14.8 | 0.91 |

| September | 12.5 | 0.93 | 17.9 | 0.91 | 14.9 | 0.91 |
|-----------|------|------|------|------|------|------|
| October   | 11.9 | 0.94 | 20.8 | 0.88 | 13.1 | 0.93 |
| November  | 12.0 | 0.94 | 27.9 | 0.80 | 12.8 | 0.93 |
| December  | 11.7 | 0.94 | 18.9 | 0.89 | 13.9 | 0.92 |
