# Peer review of "A global monthly climatology of total alkalinity: a neural network approach"

_Earth System Science Data, 2018_

## Referee Comment (RC1) · Anonymous Referee #1 · 10 Dec 2018

The authors of this study firstly extracted the relations between alkalinity and other variable (Salinity, DO, nutrients, depth, temperature and location), and adopted this relationship to generate a monthly climatology of total alkalinity. I am glad to see the a more precise alkalinity climatology dataset. However, the manuscript is poorly structured, inadequate illustrated with a lot of vague expression. There are tons of sloppy description, and the grammar is so poor that I have difficulty to understand the science. There are quite a few things that need clarification from the authors. They are listed from the most to least concerning:

1). I fail to see why "3.3 subsurface layer hypothesis" is included in this manuscript. This section plagued with serious issues: without enough background, it is very difficult for readers to understand the motivation: 1). What is the subsurface layer hypothesis? 2)

[Figure]

Whether your finding support or reject the hypothesis? 3) How this part related to the topic of monthly climatology at all? Figure 7 is also not well explained either. I have no idea how to read it. Can you add an autocorrelation figure to show the similarity between surface winter condition and subsurface layer?

2). Lines 315-390, The authors found that climatology of TA is highly dependent on the inputs. However, the logic can be improved. I would suggest re-organizing this session as this: 1). Explain the available TA climatologies, and what is the difference among them; 2) Why did authors choose WOA13 as input at last? 3). Show the monthly climatology of TA caluculate based on WOA13, discuss its variability, and compare yours and others. I also have a question related to input climatology choosing: the authors showed the difference between "AT NN WOA13"and "AT NNBATS inputs" in Fig. 10. Have the authors tried to use climatology reported by Lauvset et al 2016 as inputs? And what is the result?

3). There are a lot of unclear pronoun references across the manuscript, which make sentences very confusing and difficult to understand. There is no way to point out all of them. Please check through the Manuscript.

Specific and minor comments to the text: 1). The caption for tables should be put on the top. It is very confused with the current format.

2). Line 46, increase in temperature and ocean deoxygenation.

3). Lines 49-50, It should be five variables if including carbonate saturation state ($\Omega$). I would also suggest adding alkalinity definition before discussing its physical meaning and processes that can impact its distribution.

4). There are a lot of upwelling studies in Californian Upwelling Systems, some references may be needed here.

5). Lines 62-63 "For example, phytoplankton blooms (i.e., primary production), and the seasonality in upwelling and river flows " is not a sentence.

6). What do you mean by the "Storage of the anthropogenic CO2"? You mean the TA's seasonal cycle is important to the anthropogenic CO2 storage?

7). Lines 165-166, It is a very confused sentence. I still cannot figure out how the authors do the training.

8). Lines 192-193, again, it is a very wordy sentence, and I have no idea how the authors concluded the content after Thus.

9). Line 213. "They make up 6.5% of all the samples in this zone and 85% of them belong to the upper 100m of the water column (Table S2) ". What does the "They" represent? How did you get 6.5% based on Table S1? Adding all the %relative over n is 5.35+6.90=12.25%. I would assume the second number should be 1317/(296+1289)=83.09%. By the way, the fifth column number was also miscalculated.

10). Line 214. "in this layer of this area. . . . ..14% of the total". Please specify what "this layer of this area" represent. And what the "total" here is? The same problem with line 216 "in this area".

11). Lines 219-220. Do not know what author want to say.

12). Lines 242-244. This sentence needs to be revised. The current description makes the reader think the Lee et al (2006) have the lowest RMSE comparing to other methods. Also, Line 245. It is not "We", it should be the "NN approach".

13). Line 251. "The zones defined in the Arctic have higher RMSEs in the two studies" I have no idea what the authors want to say.

14). Lines 256-257, is not related to this section. Monthly climatology should be discussed in next section.

15). Lines 257-264, this paragraph should be put after Line 275.

16). Line 278, the authors should list the three time-series first. The same as Line 360.

Have no idea what the "other climatologies" before jumping into figures. Line 279-280, why?

17). Line 284, "We obtained similar values of RMSE of 6 $\mu$mol kg-1 and 5.5 $\mu$mol kg-1 respectively ". At which time series stations? Both values cannot be found in Table 4.

18). Lines 286-288, Too much repeat.

19). The way to mark panel is very confusing in Figure 6. Please assign each panel an ID. By the way, please explain how did you get the AT,residue without measured value in both time series stations.

20). Lines 350-359, the figure across this paragraph should be figure 10! This paragraph and following paragraph is very sloppy written. The authors should re-write it. For example, Line 355 can be simply written as "the comparisons are better (and show how better) when AT was obtained by NN with measured value as inputs". Line 356. "The differences of the two comparisons show the differences in the input variables". Have no idea what the second "differences" means. Line 360. Replace "similar" with "close". What is the "one predictor" in Line 362? What does DIVA represent? Line 363. "Furthermore, the coarser grid in the Takahashi et al. (2014) climatology involves a change of grid for the comparisons which may enhance dissimilarities". I have no idea what the authors want to say at all! Again, the above questions are only a few examples. the authors have to check through the entire MS and do the corresponding revision.

21). Line 370. "The spatial patterns of the differences between in annual mean surface AT between our and the three other climatologies under consideration are not correlated." Get lost again.

22). Lines 374-375. "It shows how the different parametrizations of the AT diverge highly at low salinities." How do the authors get this conclusion?

23). Again, what is the "the difference results"?! Add Figure 11 at end of Line 384 (. . .

of the WOA13 data. Line 387. Do you mean "below 250 m" by "in these layers" ?

24). Lines 389-390. "to be consistent, it is recommended to use the AT climatology corresponding with the other inputs used in the studies that arise from these products. " Have difficulty to understand it too.

---

## Referee Comment (RC2) · Anonymous Referee #2 · 18 Dec 2018

Recommendation: Reconsider after major revisions.

Summary: It is clear that hard and good work was done in getting this paper this far and the authors should be congratulated on their progress toward what, to me, looks like 2 papers. However, more must be done before this will be concise, clear, complete, and novel enough to warrant being broadly read by the oceanographic community.

This manuscript attempts several things:

First, it justifies the need for at TA climatology. Next it produces a new neural network for calculating TA from other seawater measurements. It then assesses the neural network and discusses likely sources of error for the method, paying special attention to riverine influences in the Arctic. Finally, it presents the climatology and compares it

to other TA climatologies in the literature.

Unfortunately, the paper uses a lot of text to only do half of the job with each of these objectives. I was left confused what use the authors had in mind for the climatology (I don't dispute that uses exist, but rather suggest that the uses were not clearly communicated). The methods used to create the neural network are similarly incomplete, where significant text is devoted to their description but not enough text is devoted to the explanation for it to make sense to people who don't already understand the material. The neural network is created, but there is only an effort to test the optimal number of neurons, and insufficient efforts are made to optimize other aspects of the NN, such as the combination of predictors used to calculate TA. The neural network assessment is incomplete (see below), and insufficient effort is also made toward comparing the new neural network to options in the literature, e.g. the Sauzude et al. CANYON reference, the Carter LIAR et al. reference (updated here: https://aslopubs.onlinelibrary.wiley.com/doi/full/10.1002/lom3.10232), or the recent Bittig et al. CONTENT methods (https://meetingorganizer.copernicus.org/EGU2018/EGU2018-2774.pdf). I attempted to do some of these comparisons on my end, but couldn't get the code to work. Finally, the presentation of the climatology itself is rushed and contains too many vague and general statements.

Going forward, consider splitting the paper into two, both halves of which will require more work before being ready for publication. Alternately, shift focus towards or away from algorithm development. If towards, then do a complete job of optimizing parameters and testing the NN against alternatives. If away, then simply omit the results from this new algorithm and use existing ones or present the climatology alongside estimates from alternatives.

If split. . .

For the first paper, a more complete case must be made as to why the new methods are

better or better specifically for generating a climatology than existing methods. Most of this case can be made by showing the new method has decreased (or comparable but independent) errors to alternatives, and this can be shown by improving the validation text with a number of new quantitative comparisons employing the various methods. Randomly selecting testing/validation data is not useful since there are large systematic TA errors along hydrographic sections. This means you will always underestimate error along a section if you train your routine with data from the same section as the test data. For this reason, CANYON authors reserved entire regions for their test data and LIAR authors omitted entire sections at a time during testing. This may or may not end up making the computed RMSE worse, but it is an important step regardless so readers know what to expect from the algorithm when they deploy it in areas where there weren't nearby dense measurements collected at the time of the estimate. It wasn't clear what omitting data with a >3 umol/kg bias for the 2nd NN training was intended to show. Uncertainty is necessary for these estimates.

For the second paper, creating a climatology from an algorithm is fast. The second paper will be complete when the authors have answered the following questions: 0. Why is a TA climatology needed? 1. What does climatological TA distribution look like? 2. What processes make it look that way? 3. What does TA variability look like? 4. What processes make it vary like that? 5. How large are the uncertainties in the climatological values, and how does this uncertainty vary regionally and with depth? And finally, 6. How do we know the answers to these questions? The current paper begins to answer all of these questions, but ultimately falls back on too many qualitative and vague statements. It therefore ends up neither concise nor complete.

Specific comments:

L40. "The capacity of the ocean..." this sentence doesn't make sense. Are you suggesting the atmospheric pCO2 would today be 520 ppm without the ocean CO2 storage? This estimate is incorrect if so.

[Figure]

L51: This definition sounds closer to the Revelle factor definition.

L53: "Processes that change salinity…" it would be better to name those processes since one can imagine processes that change salinity without changing TA.

L61: Hydrothermal TA inputs should perhaps also be mentioned.

L66: "Therefore, the knowledge of AT variability over the global oceans at monthly timescales is very useful to increase the understanding of the ocean carbon cycle and to make assessments and projections related to ocean acidification with greater rigor." Build on this. What applications specifically do you have in mind for this climatology? Why use the climatology instead of an algorithm?

L88: Why? Why is it necessary to have a seasonal climatology of subsurface TA? How deep does seasonality affect TA, and how do you know this? See: line 190.

L103-L122: This is in an unhappy medium of detail… too little detail to make any sense, and too much for the reader to quickly or confidently skip this. Either reference a paper that explains the method or fully explain it. My preference would be to fully explain the method, but move the text to a supplement so it doesn't interrupt the paper with too much detail… more detail in the text or a supplement would be necessary for the first paper if you split the paper into 2.

L140: Did you include calculated TA? TA where GLODAPv2 did not QC the data?

L159: "We kept…" I don't know what this sentence means.

L162: What is the difference between testing and validation data sets? It's possible I missed the explanatory text, but consider trying to make that a bit clearer.

L168: Explain your rationale here. It is unclear why this test would find places where the network is unable to obtain accurate values.

What does it mean to for an individual data point to have a RMSE of >3?

By my best guess, this is saying that the version of the NN that includes the data with a >3 absolute offset does better at fitting the data with a "less than 3" absolute offset than the version that only includes the "less than 3" data? The RMSE of the data with a "less than 3" offset, by definition, must be less than 3, and yet it climbs to 5.1 when you omit this data... so why bother with this analysis? Why not simply say "if we omit data with large errors our RMSE becomes small." Which do you recommend users adopt? Where and why?

L171: What does it mean to "illuminate the complexity" of a neural network? Be more specific.

L208: Random division of the datasets is inadequate for your test. See main points.

L210: How does "The samples with residuals beyond $\pm$3RMSE are 1% of the global dataset..." align with "99% of the GLODAPv2 dataset used was modelled by the network with a root-mean-squared error (RMSE) of 5.1 $\mu$mol kg-1." I'm guessing this is referring to the 2nd NN, but I was confused for a long time before this statement started to make sense.

L228: This sentence doesn't make sense to me... I suspect "disengage" is an incorrect word, but I'm not sure.

L235: It's not always riverine TA that is the problem... often it is rivers with little or no TA that dilute seawater TA in a way that is distinct from the mixing patterns in the open ocean.

L235: this paragraph presents a weak argument against removing the region... fortunately, the argument for omitting that region was not made... omit this text, but instead estimate uncertainty regionally more quantitatively.

L251: "The zones defined.." this sentence is vague.

L252: considerably

L261: what is meant by this sentence? Clarify.

l269: what is meant by this sentence? Clarify.

L273: S and T collectively provide information regarding interior ocean density and mixing patterns, which are important for predicting TA… it is not clear what you are suggesting about the link between T and CaCO3.

L278: "The bias is relatively low in the three time-series with the highest number of data. The AT computed by the NN at KNOT and K2 is higher than the measured one. Summed to the previous test, this independent test 280 with a seasonal time resolution shows the good generalization of the NN." The first two sentences here are difficult to understand and the last one does not make sense to me.

L286: This logic does not follow.

L302: This sentence does not seem to fit with the rest of the paragraph.

L310-L313: I don't understand these sentences.

L310: intra-annual would be clearer as "seasonal"

L340-342: I don't understand the logic.

L354: Isn't that figure 10?

L363: Why a change of grid for a time series?

L380: what is meant by a "continuity" in the differences?

L393: I couldn't get the code to work… some info… • Tried with the directory with NN as the active directory o As well as just with the directory with NN on the Matlab path • Tried entering within a script and on the command line • Tried with Matlab r2014b with the NN toolbox (also 2018b, but without the NN toolbox) • With inputs as single or double precision numbers • Entered: o AT_values=Neural_network_object(data_inputs); o AT_values=NN(data_inputs); o

AT_values=NN_w3RMSE(data_inputs); Invariably got the response: "Undefined function 'NNw3RMSE'/'NN'/'Neural_network_object' for input arguments of type 'single'/'double'."

I recommend making the instructions a bit more clear. I also recommend adding an example calculation so users can be sure they are getting the expected answers. You'll know you are done when the coauthors can use the function reliably without additional guidelines.

Figure 7 and elsewhere: Uppercase theta should be reserved for conservative temperature. . . use lowercase

Figure 8 and elsewhere: The font is too small.

Figure 10, right panel: How is the RMSE smaller than the bias? Clarify what is being shown here if there is a good reason.

---

## Referee Comment (RC3) · Anonymous Referee #3 · 19 Dec 2018

The authors describe a neural network approach to derive an algorithm to estimate AT from concurrent Lat/Lon, depth, T/S, oxygen and nutrient (nitrate, silicate and phosphate) inputs, based on GLODAPv2 data. They use this approach with monthly climatological fields from WOA13 to establish a global, depth-resolved, monthly AT climatology. The manuscript is clearly-structured and well-written.

I see three critical points, (1) the neural network topology selection and the second round of neural network training without control for overfitting, (2) the adequate representation of (surface) seasonality in the training data, by the neural networks, and the derived monthly climatology, and (3) the placement and comparison with other recent work on AT estimation based on GLODAPv2-trained algorithms.

I therefore suggest major revisions to the manuscript.

[Figure]

Major points:

(1a) The authors describe training of their neural networks in general terms, however, some important details remain missing.

- The selection of the best performing neural network appears subjective and is not made transparent. This needs to be improved.
  E.g., l.161: What criterion has been used to assess "best generalization in the initial testing dataset"?;
  l. 204f: 128 neurons kind of fall from the sky. Figure S1 would probably be more instructive to show RMSE for training and testing set vs. number of neurons, to make the authors' reasoning more transparent.

- Do the authors use weight regularization of the network weights? I presume so, at least for the Levenberg-Marquardt variants but probably also for their Bayesian regularization. This should be stated. It should be stated as well how the regularization (hyper-)parameter/weight was chosen (i.e., the balance between data accuracy/loss and weight penalty/loss terms in the cost function; or in other words the balance between accuracy and generalization behavior within the given network topology).

- What exactly is meant by Bayesian Regularization (l. 141 with reference to MacKay, 1992)? Please be more explicit here.
  If you used a certain, e.g., Matlab implementation/toolbox, make reference to it. MacKay (1992) describes at least three levels of Bayesian treatment, from (I) finding the 'best' (most-probable) set of weight parameters including their regularization (i.e., preserving generalization behavior by avoiding too specialized weight distributions) through (II) finding the 'best' hyperparameter values (i.e., objectively assigning the balance between data loss and complexity/regularization) to (III) model comparison (e.g., quantitatively rank different models or neural network topologies). It seems to me that only (I) has been used here? Please clarify.

Also note that, if implemented correctly (!), Bayesian regularization doesn't need cross-validation like, e.g., a backpropagation Levenberg-Marquardt learning scheme.

(1b) I think the two-step training of the networks with elimination of the testing data must be avoided (with a backpropagation/LM algorithm). Optimization of the network's parameters doesn't stop after training with the 70/15/15 % training/validation/testing data set. It continues well throughout the 80/20/0 % step, where the authors no longer have control over or means to assess overfitting. The authors' conclusion (l.165) is invalid. Given that, e.g., "[the authors] find no improvement by increasing the amount of data points in the training set" (l. 207), I don't see the point in making this questionable second step. Instead, this re-optimization of weights without control for overfitting makes the method vulnerable. It should be removed thus closing this open flank without loss in performance.

I do see the NNw3RMSE run critical, too. In essence, the authors level out areas of the ocean with higher-than-average variability (>3* global-mean-RMSE *samples* are removed, i.e., only the subset of samples that fit to the mean in these areas is retained). They do this to "improve the network mapping in the other areas" (l. 169). This *spatial* difference/distinction should be captured by the sampling position input (Lat, sLon, cLon, Depth), shouldn't it? I would argue that the (small) improvements they see in certain subregions between the NN and the NNw3RMSE runs is only due to a different local minimum found during neural network training of the one neural network selected for NN vs. the one neural network selected for NNw3RMSE, and not thanks to the omission of data in an at most adjacent or even unrelated ocean region (e.g., Equatorial Upwelling Pacific, while most samples >3*global-mean-RMSE are found at high latitudes/North Sea). Again, given that "the difference in the weighted RMSE of the two networks [NN and NNw3RMSE] is not significant" (l. 247; l.254) and that the authors consider NN the best candidate for users (l. 263), I'd suggest to drop the NNw3RMSE network.

(2) I think it is courageous to derive a monthly-resolved product from GLODAPv2 data, which in many ocean regions is far from being monthly yet seasonally-resolved. This needs further elaboration and the seasonal character needs to be demonstrated clearly.

To tackle the scarcity of winter time observations, the authors state that the lack of surface information during winter can be circumvented by using spring time observations of subsurface waters that retain the winter water signature, illustrated by figure 7. (l. 187-194).
Fair enough, but this information doesn't tell the neural network to learn it that way nor does it imply by any means that the neural network recognizes this connection. Even if winter water properties are similar between spring subsurface and winter surface samples (as in the climatological WOA13 data of figure 7), the vertical sampling location (Depth) is still different, thus ending up in a different area of the neural network input data space - giving potentially very different AT output.

The first step to convince me of this 'seasonal winter gap filling' would be to add the predicted surface and subsurface AT to figure 7 - which should approach each other during winter like the water properties.

A second step would be to give better quantification of the seasonal cycle where possible. This is probably limited to the time series stations and the North Atlantic. If the training data are seasonally well-resolved and the neural network training picks up this seasonality adequately, the seasonal cycle's amplitude from the NN (and measured inputs) should be of the same magnitude as the observed seasonal cycle's amplitude. If the training data do not reflect full seasonality, the NN tend to underestimate the seasonal cycle - with a flat line as the extreme. Such a comparison should complement the for now only qualitative assessments (e.g., figure 5).

Moreover, the (sub-)polar North Atlantic should be added as region for the sub-surface hypothesis due to the high interest in the carbon cycle in this area.

(3) Since the publication of the GLODAPv2 data set, there have been other works that use the data compilation to establish algorithms for AT estimation. Two of them are mentioned (LIARv1, Carter et al. 2016, and CANYON, Sauzede et al. 2017), however, the manuscript falls short on setting their own work into perspective of the state-of-the-art published literature.

(a) Both methods mentioned have received updates (LIRv2, Carter et al. 2018, and CANYON-B, Bittig et al. 2018), to which the comparison of the present work should be made.
Both updated algorithms are publicly available as Matlab code and use overlapping (but fewer) inputs as the authors' approach, i.e., there is no obstacle to apply them to any of the authors' data.
(b) The authors already do a decent job in assessing their work with surface-only climatologies (e.g., Lee et al. 2006), but the authors need to demonstrate more clearly how the present work improves / compares with existing, global, depth-resolved algorithms of AT (e.g., see above).
E.g., in terms of accuracy on all their time series data, not just HOT (section 3.2), surface seasonal amplitude (see point 2 above), complexity in terms of input data requirements, etc.
Interestingly, the authors don't use the year day as input either (same as LIR and CANYON-B), and nonetheless get good surface seasonality.

This point (3) is important to improve, since it will give the authors the argument of why use their algorithm (or one of the others) to derive an AT climatology from WOA13 fields, which is the main subject of this work (following the title).

Minor points:

l. 60: remove oxygen. Nutrient changes contribute to a change in AT, oxygen itself does not contribute to the charge/acid/base balance.

l. 105: The number of neurons in the output layer is adjustable? It seems to be n=1 for just AT, isn't it?

l. 124: "as previously described" - not yet done, remove.

l. 138 and 139: Which spurious oxygen value was removed / Where can it be found? (To allow reproduction by others.); Name the ocean time-series or give their GLO-DAPv2 cruise IDs.

l. 238: "As an argument ... areas." Unclear.

l. 273: Depth is rather associated as vertical sampling position.

l. 279: Any ideas why there is such a bias? Should be commented.

---

## Author Comment (AC1) · 26 Feb 2019

**RC**: Referee comment; **AR**: author's response; **AC**: author's changes in manuscript.

**Referee #1 response**

**RC**: The authors of this study firstly extracted the relations between alkalinity and other variable (Salinity, DO, nutrients, depth, temperature and location), and adopted this relationship to generate a monthly climatology of total alkalinity. I am glad to see the a more precise alkalinity climatology dataset. However, the manuscript is poorly structured, inadequate illustrated with a lot of vague expression. There are tons of sloppy description, and the grammar is so poor that I have difficulty to understand the science. There are quite a few things that need clarification from the authors. They are listed from the most to least concerning.

**AR**: We are pleased to see that you appreciate the more precision of our alkalinity climatology dataset as well as we appreciate all your useful comments. Thank you so much for the thorough revision of the manuscript. We hope to improve it based on all the comments of the 3 referees. In order to maintain a balance between your comment about the poor structure and the vague expression of the manuscript and the comment of the Referee #3: "The manuscript is clearly-structured and well-written", we have reviewed these points which are mainly depicted in the next responses. At the end of the answers is attached the new version of the manuscript for a global view.

**RC**: I fail to see why "3.3 subsurface layer hypothesis" is included in this manuscript. This section plagued with serious issues: without enough background, it is very difficult for readers to understand the motivation: 1). What is the subsurface layer hypothesis? 2) Whether your finding support or reject the hypothesis? 3) How this part related to the topic of monthly climatology at all? Figure 7 is also not well explained either. I have no idea how to read it. Can you add an autocorrelation figure to show the similarity between surface winter condition and subsurface layer?

**AR**: The motivation was explained in lines 185-194. In brief, the motivation to test this hypothesis, as Vázquez-Rodríguez et al. (2012) proposed, has been the lack of winter data in some regions. In section 3.3, we demonstrated in the same way as Vázquez-Rodríguez et al. (2012) did, that their hypothesis is verified for other regions with a lack of data in

GLODAPv2. For the non-winter months when samples were taken from the subsurface layer similar relations exist between predictors and AT. This fact avoids a seasonal bias in these areas where the presence of non-winter data could bias the climatology to the months represented in the training dataset. Figure 7 is exactly designed and described as Figure 2 in Vázquez-Rodriguez et al. (2012). The reader is referred for further detail to the cited paper for an in-depth analysis of the similar results that are shown in their Figure 2.

We have added an additional analysis in this section to reinforce the subsurface hypothesis through training a neural network without winter data and testing its ability to fit the independent winter dataset. That is, extract the relations only from non-winter data and test them in winter independent data.

**AC**: In methodology: "GLODAPv2 contains quality controlled measurements in all ocean basins from the 1970s until 2013 (Olsen et al., 2016). However, winter data are scarce to absent in some high latitude regions because adverse weather conditions prevents field activities in that season (Fig. 3). In surface ocean, this temporal bias can be avoided with the help of the subsurface data from seasons with sufficient samples. Vázquez-Rodríguez et al. (2012) demonstrated how the subsurface ocean layer in the Atlantic Ocean can retain the footprint of the water mass formation from the preceding winter in the following months and, therefore, of the surface conditions. The winter relationship between inputs and AT needed to produce an all-season surface climatology are mostly preserved in this subsurface layer. The validity of this hypothesis was tested in other regions (Fig. 3) following Vázquez-Rodríguez et al. (2012). These areas were chosen based on the non-availability of AT data in two or more consecutive months in the same oceanographic regime as the colored area in Fig. 3.

To reinforce the previous test and to assess the ability of the neural network in overcoming the lack of winter data in other depths, a neural network was trained excluding all winter data in GLODAPv2 (GLODAPv2_nowinter) and tested in the excluded and independent winter dataset (GLODAPv2_winter). The procedure to create and to train the network was the same as described previously."

In results and discussion: "We found that the optimal depth range of the subsurface layer defined by Vázquez-Rodríguez et al. (2012) for the North Atlantic Ocean (100-200 m) must be modified in other regions. In the area analyzed in the Indian Ocean (Fig. 3), the

subsurface layer hypothesis is verified in the same depth range of that study. However, the other areas (Fig. 3) show that the range of the subsurface layer is in the range of 50-100 m. The different strengths of deep mixing and convection in winter could explain this fact.

The properties analyzed in the four areas defined in Fig. 3 show, as expected, a higher monthly variability in the ocean surface than in the subsurface layers. The seasonal variability depicted in Fig. 8 will likely be typical of a larger region within a similar oceanographic regime for each defined area. The surface winter conditions of the analyzed properties are quite similar to those in the subsurface layer during, at least, one of the four consecutive months following winter in all areas (Fig. 8).

The optimal number of neurons in the network trained with GLODAPv2_nowinter dataset to reinforce the subsurface layer hypothesis and to assess the layers below surface ocean was 100. The reduction of the number of neurons compared to the previous networks was because this new dataset contains less data. Thus, maintaining or increasing the number of neurons would produce overfitting. This new network provides statistics in the GLODAPv2_nowinter dataset similar to those of the network used to create the climatology (NNGv2) in GLODAPv2 dataset (Table 1 vs Table 7). But, of greater importance are the statistics resulted from the GLODAPv2_winter dataset (Table 7) which reinforce the subsurface layer hypothesis. The low error reached in this independent winter dataset shows how the network is able to obtain the winter relations in any depth from the function fitted with data from other seasons. Therefore, the lack of winter data in different regions does not automatically mean that the climatology will be biased towards the more sampled seasons."

**RC**: Lines 315-390, The authors found that climatology of TA is highly dependent on the inputs. However, the logic can be improved. I would suggest re-organizing this session as this: 1). Explain the available TA climatologies, and what is the difference among them; 2) Why did authors choose WOA13 as input at last? 3). Show the monthly climatology of TA caluculate based on WOA13, discuss its variability, and compare yours and others.

I also have a question related to input climatology choosing: the authors showed the difference between "AT NN WOA13"and "AT NNBATS inputs" in Fig. 10. Have the

authors tried to use climatology reported by Lauvset et al 2016 as inputs? And what is the result?

**AR**: The sequence for the referred lines is: 1) To show the climatology and discuss the patterns and the variability obtained; 2) To analyze if the variability is coherent comparing the climatology with the unique climatological measured data (BATS and HOT time-series); 3) Once we have shown that the climatology is robust, we have compared all the available climatologies and ours. Your point 1) and the last sentence of the 3) is our point 3). We have considered your suggestion but separating the comparisons will break the logic. We have used the climatologies of our input variables given by Lauvset et al. (2016) to assess the difference in both methods and inputs between their climatology and ours. As the climatologies of Lauvset et al. (2016) are not seasonal, the comparison of the Fig. 10 that you suggest cannot be done.

**RC**: There are a lot of unclear pronoun references across the manuscript, which make sentences very confusing and difficult to understand. There is no way to point out all of them. Please check through the Manuscript.

**AR**: The whole manuscript has been checked by a native speaker of English.

**RC**: 1). The caption for tables should be put on the top. It is very confused with the current format.

**AR**: Changed.

**RC**: 2). Line 46, increase in temperature and ocean deoxygenation.

**AR**: Added.

**RC**: 3). Lines 49-50, It should be five variables if including carbonate saturation state ($\Omega$). I would also suggest adding alkalinity definition before discussing its physical meaning and processes that can impact its distribution.

**AR**: Added.

**AC**: Dickson (1981) defined $A_T$ as:

$A_T = [HCO_3^-] + 2[CO_3^{2-}] + [B(OH)_4^-] + [OH^-] + [HPO_4^{2-}] + 2[PO_4^{3-}] + [SiO(OH)_3^-]$

$+ [HS^-] + 2[S^{2-}] + [NH_3] - [H^+] - [HSO_4^-] - [HF] - [H_3PO_4]$

**RC**: 4). There are a lot of upwelling studies in Californian Upwelling Systems, some references may be needed here.

**AR**: Added Millero et al. (1998).

**RC**: 5). Lines 62-63 "For example, phytoplankton blooms (i.e., primary production), and the seasonality in upwelling and river flows" is not a sentence.

**AR**: Changed

**AC**: "Phytoplankton blooms (i.e., primary production) and the seasonality in upwelling and river flows are some of the more remarkable processes associated with the time variability of AT."

**RC**: 6). What do you mean by the "Storage of the anthropogenic CO2"? You mean the TA's seasonal cycle is important to the anthropogenic CO2 storage?

**AR**: We have added a reference to the paper Renforth and Henderson (2017) about carbon sequestration in the ocean where the seasonality of $A_T$ is mentioned.

Renforth, P., and G. Henderson: Assessing ocean alkalinity for carbon sequestration, Rev. Geophys., 55, 636–674, doi:10.1002/2016RG000533, 2017.

**RC**: 7). Lines 165-166, It is a very confused sentence. I still cannot figure out how the authors do the training.

**AR**: Text between lines 151 and 166 has been deleted and new one was added to clarify the training procedure. A figure was also added (Fig. 2).

**AC**: New text: "The training procedure was carried out in MATLAB. We tested 16, 32, 64, 128 and 264 neurons in the hidden layer based on the results of Velo et al. (2013). For each number of neurons, we trained 10 networks always using the same 90% of GLODAPv2 for training (Fig. 2, Static level). The remaining 10% was used as a static test (Fig. 2, Static level). Both subsets contained samples randomly distributed in the ocean to evaluate the maximum possible relationships between the input variables and AT through all oceanographic regimes, that is, to capture most of the variability in all the variables and not restricting the sets to specific areas. Each of the 10 networks starts the training procedure with random weight and bias values and a random division of the training static dataset into three portions: 70% for training, 15% for testing and 15% for validation (Fig. 2, Dynamic level). These differences make minimization of the cost function different for each network due to the complexity of the weight-error space and, consequently, their different starting points in that space. As each network is different, keeping static sets allows one to determine which network best generalizes in the same test set. The selected network is the one that produces the lowest RMSE in the training data (validation + training dynamic) and in the test data (static + dynamic), considering a non-significant difference between both RMSEs to prevent overfitting. The network derived from this process will be referred as NNGv2.

Once we found an adequate network configuration, we increased the amount of data in the training dynamic set to capture more relations between the inputs and AT. The new percentages of the dynamic sets were: 80% training, 20% validation and 0% testing. The latter set is only necessary to compare different models and is not used during the training. However, the static test set was held to evaluate the generalization of each of the 10 networks to select the best one."

[Figure]

**Figure 2. Division of the data for the training of the network. The data in the sets of the static level is the same for all the networks to train. The data in the sets of the dynamic level is randomly selected for each network to train.**

**RC**: 8). Lines 192-193, again, it is a very wordy sentence, and I have no idea how the authors concluded the content after Thus.

**AR**: Vázquez-Rodríguez et al. (2012) concluded that the relations between some variables and AT in wintertime at surface can be found in other seasons in a subsurface layer. Therefore, we are using their conclusion and adapt it to our study. Furthermore, we test the validity of their hypothesis in other ocean areas.

**RC**: 9). Line 213. "They make up 6.5% of all the samples in this zone and 85% of them belong to the upper 100m of the water column (Table S2)". What does the "They" represent? How did you get 6.5% based on Table S1? Adding all the %relative over n is 5.35+6.90=12.25%. I would assume the second number should be 1317/(296+1289)=83.09%. By the way, the fifth column number was also miscalculated.

**AR**: "They" is used to refer to the samples with residuals beyond +-3RMSE in this paragraph because we did not want to repeat "samples with residuals beyond +-3RMSE" all the time.

6.5% is the percentage of samples with residuals beyond +-3RMSE over total GLODAPv2 samples in the indicated area: latitudes greater than 60ºN. From table S2: (296+1289)/(5531+18684)=0.065.

About the second number, 85%, your assumption is correct. We wrote the value obtained with a previous network that we tested. We have changed the number in the text and in Table S2. The percentage in the fifth column was also updated.

**AC**: The paragraph between lines 210 and 216 has been changed for a better understanding as follows: "Samples with residuals beyond ±3RMSE are 1% of the GLODAPv2 dataset. The spatial distribution of these samples (Fig. S3) show that they are confined to certain areas, mainly in the ocean surface (Fig. 4). Most are in the Northern Hemisphere (Fig. S3 and Fig. 4). Specifically, 64% are from latitudes north of 60ºN (Table S1). In this area, 6.5% of GLODAPv2 samples have residuals beyond ±3RMSE and 83.1% of these samples are from the upper 100m (Table S2). In these depth and latitude ranges, the samples with high residuals make up 14% of the GLODAPv2 samples here and they typically have salinities lower than 34 (Table S3; Fig. S3). A monthly

analysis in the previously indicated ranges shows that the largest number of samples with residuals beyond ±3RMSE are from the summer months. About 15-19% of all the samples from this season in this area have residuals higher than ±3RMSE (Table S4)."

**RC**: 10). Line 214. "in this layer of this area: : :: : :14% of the total". Please specify what "this layer of this area" represent. And what the "total" here is? The same problem with line 216 "in this area".

**AR**: Changed.

**AC**: "in this layer of this area" → "In these depth and latitude ranges"

"total" → "GLODAPv2 samples here"

**RC**: 11). Lines 219-220. Do not know what author want to say.

**AR**: Because of the peculiar process in the Arctic, the subsurface layer hypothesis might not be valid here and the climatology could represent only the characteristic AT of the sampled months.

**RC**: 12). Lines 242-244. This sentence needs to be revised. The current description makes the reader think the Lee et al (2006) have the lowest RMSE comparing to other methods. Also, Line 245. It is not "We", it should be the "NN approach".

**AR**: This is indeed that we want to say. Compared with other methods on the generation of a monthly climatology, as it is specified in the previous sentence: "previous studies on generation of monthly climatologies. Anyway, we have changed some things in the paragraph for a better understanding, including the suggested one in your comment.

**AC**: New text: "In the global ocean surface layer, the RMSE obtained with the neural network approach is lower than that obtained by previous studies on generation of monthly climatologies (Table 2 and 3). In the past, relationships between SST and SSS with AT by Lee et al. (2006) have been shown to produce the lowest RMSE (area-weighted RMSE of 8.1 µmol kg-1) in the AT computation to create a monthly climatology. However, applying the relations of that study to GLODAPv2, the obtained

weighted RMSE is higher than the one from the neural network (Table 2). Neural network approach obtained a better fit in all the areas defined in the study of Lee et al. (2006) (Table 2). NN±3RMSE improves the results obtained with the NNGv2 in almost all the regions, being the most remarkable the Equatorial Upwelling Pacific. However, the difference in the weighted RMSE of the two networks is not significant."

**RC**: 13). Line 251. "The zones defined in the Arctic have higher RMSEs in the two studies" I have no idea what the authors want to say.

**AR**: Changed.

**AC**: New text: "The AT computed in the zones defined in the Arctic have higher RMSEs in the two approaches (Takahashi et al. (2014) and this study; Table 3)."

**RC**: 14). Lines 256-257, is not related to this section. Monthly climatology should be discussed in next section.

**AR**: We are comparing the accuracy of the available methods designed to create a monthly climatology. Therefore, the monthly climatology is not discussed here, only the fitting techniques.

**RC**: 15). Lines 257-264, this paragraph should be put after Line 275.

**AR**: In this paragraph we are showing which of our two networks is selected to create the climatology depending on what is discussed in the previous paragraph. Therefore, it follows a logic sequence and put it after Line 275 would break the sequence of the text.

**RC**: 16). Line 278, the authors should list the three time-series first.

**AR**: Added.

**RC**: The same as Line 360. Have no idea what the "other climatologies" before jumping into figures.

**AR**: Changed

**AC**: "Compared to the other climatologies" → "Compared to other climatologies (Lee et al. (2006), Takahashi et al. (2014) and Lauvset et al. (2014))"

**RC**: Line 279-280, why?

**AR**: To specify the good performance of the network in datasets different from the previously tested (GLODAPv2), mainly in the time resolution, one of the important features of our study. To sum up, previously we showed the generalization of the network in GLODAPv2 samples located randomly around the world and now we are showing the generalization in specific locations over samples measured monthly and/or seasonally.

**RC**: 17). Line 284, "We obtained similar values of RMSE of 6 _mol kg-1 and 5.5 _mol kg-1 respectively". At which time series stations? Both values cannot be found in Table 4.

**AR**: This paragraph has been deleted and we have added a more concise comparison with LIARv2 and CANYON-B.

**AC**: "The LIARv2 and CANYON-B methods to compute $A_T$ also model the time-series data quite well (Table 6). Significant differences among the three methods are obtained in HOT and ESTOC. In HOT, NNGv2 and CANYON-B reach a better fit of $A_T$ than LIARv2 suggesting that a non-linear technique is more adequately to model $A_T$ in this area (Table 6). In ESTOC, NNGv2 and LIARv2 are the best options to model the $A_T$ variability (Table 6). Here, the $A_T$ computed with LIARv2 with the option of the free equation choice activated results in a greater election of the equations that include nutrients as predictors. This result show how in this area the inclusion of nutrients as predictors contributes to improve the model of $A_T$. Like NNGv2, both methods have a considerable bias in K2 and KNOT (Table 6) that reinforce the two reasons suggested previously."

**Table 6: RMSE and bias between measured $A_T$ and the $A_T$ computed with both LIARv2 and CANYON-B methods. The comparison was done for the same samples evaluated in Table 5.**

| Time-Series | LIARv2 | | CANYON-B | |
| --- | --- | --- | --- | --- |
| | | bias | | bias |
| | RMSE (μmol kg$^{-1}$) | (μmol kg$^{-1}$) | RMSE (μmol kg$^{-1}$) | (μmol kg$^{-1}$) |
| HOT | 6.6 | -0.6 | 5.8 | -0.6 |
| BATS | 6.3 | 0.1 | 6 | -0.4 |

| | | | | |
|---|---|---|---|---|
| ESTOC | 3.4 | 0.8 | 4.2 | 3.2 |
| KNOT | 4.8 | -6.6 | 4.5 | -7.2 |
| K2 | 3 | -3.0 | 3 | -3.3 |

**RC**: 18). Lines 286-288, Too much repeat.

**AR**: Deleted in the new version of the manuscript. See previous comment.

**RC**: 19). The way to mark panel is very confusing in Figure 6. Please assign each panel an ID. By the way, please explain how did you get the AT,residue without measured value in both time series stations.

**AR**: Figure 6 and its title have been changed. The AT residuals were obtained from the difference between measured and computed AT as is explained in the figure title (Figure 6 panels in the central column). The panels in the right column are the same as those of the central column but applying an interpolation (DIVA) (as it is written in the title of the figure) for visual purposes.

**AC**:

[Figure]

**Figure 7: Left column: Computed A_T for the upper 550m of the water column at the BATS and HOT time-series stations. Central column: Difference between measured and computed A_T. Colored dots show samples where A_T was measured. Black dots show samples where A_T was not measured but the network inputs were. Right column: Difference between measured and computed A_T interpolated with Data-Interpolating Variational Analysis (DIVA; Troupin et al., 2010). This figure was made with Ocean Data View (Schlitzer, 2016).**

**RC**: 20). Lines 350-359, the figure across this paragraph should be figure 10!

**AR**: Changed.

**RC**: This paragraph (Lines 350-359) and following paragraph is very sloppy written. The authors should re-write it. For example, Line 355 can be simply written as "the comparisons are better (and show how better) when AT was obtained by NN with measured value as inputs".

**AR**: That sentence has been rewritten following your indication. However, we don't consider including the statistics in the text. The figure is referenced in that sentence and the reader can see the statistics in the figure together with graphs for a better understanding.

**AC**: "The comparisons are better when $A_T$ is computed by NNGv2 using as inputs the measured values in the time-series (Fig. 11, purple line)"

**RC**: Line 356. "The differences of the two comparisons show the differences in the input variables". Have no idea what the second "differences" means.

**AR**: Changed

**AC**: "The differences of the two comparisons show the differences in the input variables (WOA13 climatological fields vs time-series input data)."

**RC**: Line 360. Replace "similar" with "close".

**AR**: Changed.

**RC**: What is the "one predictor" in Line 362?

**AR**: It is indicated in the introduction when the description of the study of Takahashi et al. (2014) is written. However, we have repeated it here again.

**AC**: "…one predictor…" → "…one predictor (salinity)…"

**RC**: What does DIVA represent?

**AR**: The acronym is depicted in Figure 6 title, which is previous to this line in the text sequence. However, we have added it here again.

**AC**: "DIVA (Data-Interpolating Variational Analysis)

**RC**: Line 363. "Furthermore, the coarser grid in the Takahashi et al. (2014) climatology involves a change of grid for the comparisons which may enhance dissimilarities". I have no idea what the authors want to say at all! Again, the above questions are only a few examples. the authors have to check through the entire MS and do the corresponding revision.

**AR**: To compare our climatology with the one of Takahashi et al. (2014), ours was transferred to their grid through an average which could be a contribution to the differences between the climatologies. We have rewritten the sentence.

**AC**: "Furthermore, the transfer of our climatology to the coarser grid of Takahashi et al. (2014) for the comparisons may enhance dissimilarities."

**RC**: 21). Line 370. "The spatial patterns of the differences between in annual mean surface AT between our and the three other climatologies under consideration are not correlated." Get lost again.

**AR**: The "in" is a typo.

**AC**: "the surface spatial patterns of the differences between the annual mean of our AT climatology and the three other ones under consideration are not correlated (Figure S7)."

**RC**: 22). Lines 374-375. "It shows how the different parametrizations of the AT diverge highly at low salinities." How do the authors get this conclusion?

**AR**: The previous sentence is "The largest differences in these two ocean basins are mainly located close to the river mouths.". Therefore, it is clearly shown that the different approaches to compute $A_T$ compared here (this study vs Takahashi) show the largest differences at low salinities.

**RC**: 23). Again, what is the "the difference results"?!

**AR**: Changed.

**AC**: "An important cause of the differences between the climatologies stems from the use of different inputs to generate them."

**RC**: Add Figure 11 at end of Line 384 (… of the WOA13 data).

**AR**: Added.

**RC**: Line 387. Do you mean "below 250 m" by "in these layers"?

**AR**: Yes, we do. We have joined the two sentences for a better understanding.

**AC**: "The values of the RMSE of the comparisons like those in Fig. 11 but below 250m are in the range of 4 to 6 $\mu$mol kg$^{-1}$ and the improvement caused by the inputs usage is reduced to around 1 $\mu$mol kg$^{-1}$".

**RC**: 24). Lines 389-390. "to be consistent, it is recommended to use the AT climatology corresponding with the other inputs used in the studies that arise from these products." Have difficulty to understand it too.

**AR**: Each climatology depends on the fields used to create them (Lee et al (2006): WOA01; Takahashi et al. (2014): Lauvset et al. (2016): GLODAPv2; this study: WOA13). Therefore and as an example, if an ocean carbon cycle modeling study is using the temperature fields of WOA13, the recommendation that we make is that our AT climatology should then be used too.

[revised manuscript text omitted]

---

## Author Comment (AC2) · 26 Feb 2019

**RC**: Referee comment; **AR**: author's response; **AC**: author's changes in manuscript.

**Referee #2 response**

**RC**: Summary: It is clear that hard and good work was done in getting this paper this far and the authors should be congratulated on their progress toward what, to me, looks like 2 papers. However, more must be done before this will be concise, clear, complete, and novel enough to warrant being broadly read by the oceanographic community. This manuscript attempts several things: First, it justifies the need for at TA climatology. Next it produces a new neural network for calculating TA from other seawater measurements. It then assesses the neural network and discusses likely sources of error for the method, paying special attention to riverine influences in the Arctic. Finally, it presents the climatology and compares it to other TA climatologies in the literature. Unfortunately, the paper uses a lot of text to only do half of the job with each of these objectives. I was left confused what use the authors had in mind for the climatology (I don't dispute that uses exist, but rather suggest that the uses were not clearly communicated). The methods used to create the neural network are similarly incomplete, where significant text is devoted to their description but not enough text is devoted to the explanation for it to make sense to people who don't already understand the material. The neural network is created, but there is only an effort to test the optimal number of neurons, and insufficient efforts are made to optimize other aspects of the NN, such as the combination of predictors used to calculate TA. The neural network assessment is incomplete (see below), and insufficient effort is also made toward comparing the new neural network to options in the literature, e.g. the Sauzude et al. CANYON reference, the Carter LIAR et al. reference (updated here: https://aslopubs.onlinelibrary.wiley.com/doi/full/10.1002/lom3.10232), or the recent Bittig et al. CONTENT methods (https://meetingorganizer.copernicus.org/EGU2018/EGU2018-2774.pdf). I attempted to do some of these comparisons on my end, but couldn't get the code to work. Finally, the presentation of the climatology itself is rushed and contains too many vague and general statements. Going forward, consider splitting the paper into two, both halves of which will require more work before being ready for publication. Alternately, shift focus towards or away from algorithm development. If towards, then do a complete job of optimizing parameters and testing the NN against alternatives. If away, then simply omit the results from this new algorithm and use existing ones or present the climatology alongside estimates from alternatives. If split: : : For the first paper, a more complete case must be made as to why the new methods are better or better specifically for generating a climatology than existing methods. Most of this case can be made by showing the new method has decreased (or comparable but independent) errors to alternatives, and this can be shown by improving the validation text with a number of new quantitative comparisons employing the various methods. Randomly selecting testing/validation data is not useful since there are large systematic TA errors along hydrographic sections. This means you will always underestimate error along a section

if you train your routine with data from the same section as the test data. For this reason, CANYON authors reserved entire regions for their test data and LIAR authors omitted entire sections at a time during testing. This may or may not end up making the computed RMSE worse, but it is an important step regardless so readers know what to expect from the algorithm when they deploy it in areas where there weren't nearby dense measurements collected at the time of the estimate. It wasn't clear what omitting data with a >3 umol/kg bias for the 2nd NN training was intended to show. Uncertainty is necessary for these estimates. For the second paper, creating a climatology from an algorithm is fast. The second paper will be complete when the authors have answered the following questions: 0. Why is a TA climatology needed? 1. What does climatological TA distribution look like? 2. What processes make it look that way? 3. What does TA variability look like? 4. What processes make it vary like that? 5. How large are the uncertainties in the climatological values, and how does this uncertainty vary regionally and with depth? And finally, 6. How do we know the answers to these questions? The current paper begins to answer all of these questions, but ultimately falls back on too many qualitative and vague statements. It therefore ends up neither concise nor complete.

**AR**: Thank you so much for the thorough revision of the manuscript. We hope to improve it based on all the comments of the 3 referees. At the end of the answers is attached the new version of the manuscript for a global view.

We have clarified and added some of the multiple uses of a climatology of AT.

We have modified and included more aspects about the methodology in the creation of the neural network. We have written the key aspects to understand what we are doing but is not the scope of this manuscript to deepen neural networks. Several references about neural networks are given through the text for readers who want to immerse in the neural network world. This methodology section was divided in subsections.

The selection of predictors is based on bibliography to include all the variables related with AT variability. Moreover, with the new comparison with LIARv2 and CANYON-B, that include less predictors, we demonstrate the importance to include all the possible predictors related to AT variability to compute AT with a lower error.

We have added a comparison with the newest methods in AT computation (LIARv2 and CANYON-B). The comparison was done in the GLODAPv2 dataset used in our study, in a GLODAPv2 subset of the samples where AT QC was done and in the time-series data.

We keep the random selection of the sets because we think that is more important to capture more variability in both the training and the test sets to evaluate the complete range of variability in GLODAPv2 AT. A well-trained neural network is able to avoid fitting the error in the possible systematic errors along sections. It can be seen in the new test that we have done in the section on the subsurface layer hypothesis. In this test we remove all winter data, and therefore it is similar to the suggested approach in your comment. The error in this independent set, which includes data of the two hemispheres for three months and therefore complete oceanographic sections, is quite low and of the same magnitude when computed by the two networks (the NN and the new one).

Therefore, there are no significant differences in the results between our approach and the suggested one, but in ours more variability is been captured in the relations created and evaluated in our test. Answering to your comment: "when they deploy it in areas where there weren't nearby dense measurements collected at the time of the estimate", this test also shows the potential of the network in computing AT in areas like those you refer.

**AC**: These changes are all shown in the new manuscript that we attach at the end of the comments.

**RC**: Specific comments: L40. "The capacity of the ocean: : :" this sentence doesn't make sense. Are you suggesting the atmospheric pCO2 would today be 520 ppm without the ocean CO2 storage? This estimate is incorrect if so.

**AR**: No, we are not suggesting that the atmospheric pCO2 would today be 520 ppm. The sentence was poorly written. We wanted to say that the 30% of the anthropogenic emissions were absorbed by the ocean. We have rewritten the sentence.

**AC**: "The oceanic capacity to dissolve and store atmospheric $CO_2$, and the subsequent chemical speciation, have resulted in approximately 30% less anthropogenic $CO_2$ in the atmosphere (Le Quéré et al., 2017) than it would otherwise have."

**RC**: L51: This definition sounds closer to the Revelle factor definition.

**AR**: Changed

**AC**: "AT is a key variable in the framework of ocean acidification because of what it is associated: the oceanic capacity to buffer pH changes"

**RC**: L53: "Processes that change salinity…" it would be better to name those processes since one can imagine processes that change salinity without changing TA.

**AR**: They are named in the following two sentences. "… precipitation and evaporation…", "…rivers runoff…"

**RC**: L61: Hydrothermal TA inputs should perhaps also be mentioned.

**AR**: Added.

**AC**: "Finally, hydrothermal vents could modify the concentration of $A_T$ locally (Chen, 2002).

**RC**: L66: "Therefore, the knowledge of AT variability over the global oceans at monthly timescales is very useful to increase the understanding of the ocean carbon cycle and to make assessments and projections related to ocean acidification with

greater rigor." Build on this. What applications specifically do you have in mind for this climatology? Why use the climatology instead of an algorithm?

**AR**: The main application and the motivation to design this climatology is its use in modeling studies (for example, coupling a circulation and a biogeochemical model). Therefore, a climatology is needed to have both initial and boundary conditions. To clarify the main application some was added to the end of the referred sentence.

**AC**: "A monthly $A_T$ climatology that captures most of the spatiotemporal variability can be used as initial and/or boundary conditions in biogeochemical models, in evaluating the $CaCO_3$ pump (e.g., Carter et al., 2014) or computing the ocean inventory of anthropogenic $CO_2$ (e.g., Steinfieldt et al., 2009)."

**RC**: L88: Why? Why is it necessary to have a seasonal climatology of subsurface TA? How deep does seasonality affect TA, and how do you know this? See: line 190.

**AR**: Figure S6 shows the seasonality of AT at different depths. It is reduced with depth mainly because of the low or even absent seasonality in the input variables. However, there are other variables involved in AT variability that can change and are not included directly in our approach (e.g. CaCO3 formation and dissolution) but that could be explicitly represented through other variables that we included (position: latitude, longitude and depth). Analyzing time-series at depth, some seasonality is easily detected even at high depths. Therefore, it is important to have a seasonal climatology at least to certain depths. As there is no constant AT value with time in all locations of the ocean at any depth, it is not a logical to move from a seasonal climatology to an annual from a depth to another the continuity.

About Line 190. It has not been included in the new paragraph.

**RC**: L103-L122: This is in an unhappy medium of detail… too little detail to make any sense, and too much for the reader to quickly or confidently skip this. Either reference a paper that explains the method or fully explain it. My preference would be to fully explain the method, but move the text to a supplement so it doesn't interrupt the paper with too much detail… more detail in the text or a supplement would be necessary for the first paper if you split the paper into 2.

**AR**: The referred paragraphs follow the following scheme: 1) Lines 103-107. Description of the main elements of the neural network. 2) Lines 107-109. Operation of a neuron. 3) Lines 110-117. Explanation of the training procedure with enough clarifications to understand how a neural network works. References to know the mathematics of the the Levenberg-Marquardt and the backpropagation algorithms are given here. 4) Lines 118-122. Architecture of the neural network used in this study and why we selected it.

In other studies that use neural networks to fit variables of the seawater CO2 chemistry the description is even shorter (e.g. Velo et al. (2013); Sauzède et al. (2017)). Therefore, we think that we give enough information together with the references provided to

understand what a neural network does. To go deeper, the linear algebra behind neural networks should be explained. However, the reader can find it in the given references, although it is not necessary to understand what we are doing in our work. Therefore, we don't consider splitting the manuscript nor include many more details about neural networks because it would repeat what is already written in other studies. However, we have rearranged the methodology to include some more details about the creation and an operation of a neural network as well as more references. It can be seen in the new manuscript attached at the end of the comments.

**RC**: L140: Did you include calculated TA? TA where GLODAPv2 did not QC the data?

**AR**: Yes, we did. If there is some kind of error in these samples the neural network is not going to model it. Evidence of this can be seen in the good generalization of the network as it was shown through the good results in the test set. Other evidence is in the new test we have done comparing LIARv2, CANYON-B and NN in a GLODAPv2 subset excluding the samples where the QC was not done. Here, the statistics of our NN are better than the other two methods, which did not include non-QC samples in their "trainings".

**RC**: L159: "We kept: : :" I don't know what this sentence means.

**AR**: This paragraph has been changed for a better understanding.

**AC**: "The training procedure was carried out in MATLAB. We tested 16, 32, 64, 128 and 264 neurons in the hidden layer based on the results of Velo et al. (2013). For each number of neurons, we trained 10 networks always using the same 90% of GLODAPv2 for training (Fig. 2, Static level). The remaining 10% was used as a static test (Fig. 2, Static level). Both subsets contained samples randomly distributed in the ocean to evaluate the maximum possible relationships between the input variables and AT through all oceanographic regimes, that is, to capture most of the variability in all the variables and not restricting the sets to specific areas. Each of the 10 networks starts the training procedure with random weight and bias values and a random division of the training static dataset into three portions: 70% for training, 15% for testing and 15% for validation (Fig. 2, Dynamic level). These differences make minimization of the cost function different for each network due to the complexity of the weight-error space and, consequently, their different starting points in that space. As each network is different, keeping static sets allows one to determine which network best generalizes in the same test set. The selected network is the one that produces the lowest RMSE in the training data (validation + training dynamic) and in the test data (static + dynamic), considering a non-significant difference between both RMSEs to prevent overfitting. The network derived from this process will be referred as NNGv2."

**RC**: L162: What is the difference between testing and validation data sets? It's possible I missed the explanatory text, but consider trying to make that a bit clearer.

**AR**: Validation is used to finish the training process avoiding overfitting. Testing is not used in the training per se and, therefore, it is used as an independent set to compare different models. We have rewritten the paragraph to make it clearer.

**AC**: "Two different training techniques were tested: the Levenberg-Marquardt method (lm) and the Bayesian Regularization (br) (both detailed in Hagan et al., 2014). In a similar study, Velo et al. (2013) demonstrated that these techniques give the best network performance among those they tested. Except for the number of neurons, the two algorithms were implemented with the default options of the MATLAB functions trainlm and trainbr (detailed in Beale et al., 2017). These two functions prevent overfitting in different ways. The trainlm function usually needs to be fed with the data divided in three sets: a training set to obtain the relationships between variables, a validation set to prevent overfitting and a test set to compare different networks. Here, the training was stopped when the error in the validation set increased during 6 consecutive iterations of the training process to avoid overfitting. This process is known as early stopping (Hagan et al., 2014). The final values of the network weights and biases are those reached before the first of these iterations. The trainbr function adds a regularization parameter to the cost function to make the fit smoother in order to avoid overfitting. The validation set is not present in this technique. The end of the training is based on network convergence through parameter stabilization by an automatic process known as automated Bayesian Regularization (Hagan et al., 2014; Beale et al., 2017). See Beale et al. (2017) and references therein for a detailed description of the two functions tested."

**RC**: L168: Explain your rationale here. It is unclear why this test would find places where the network is unable to obtain accurate values.

What does it mean to for an individual data point to have a RMSE of >3? By my best guess, this is saying that the version of the NN that includes the data with a >3 absolute offset does better at fitting the data with a "less than 3" absolute offset than the version that only includes the "less than 3" data? The RMSE of the data with a "less than 3" offset, by definition, must be less than 3, and yet it climbs to 5.1 when you omit this data: : : so why bother with this analysis? Why not simply say "if we omit data with large errors our RMSE becomes small." Which do you recommend users adopt? Where and why?

**AR**: The approach here comes from the hypothesis that not all the processes are captured through the inputs that we used in our network. Therefore, some regions where specific processes occur could have high errors. This is what happens, for example, in the Arctic. So, if you represent in a map the samples with differences between the measured and the computed AT (Fig. 3; or Fig. S2 to see only the samples with the highest errors), you can see that the highest errors are mainly concentrated in specific regions.

The so-called 3σ rule is widely used for outlier detection and this was the main reason for choosing the criterion here (changing the standard deviation for the RMSE). A sample with an error greater than that threshold could mean two main things: 1)

Uncertainties in the measurements of any input and/or in AT; 2) Influence of other variables not included as input in the AT variability. Areas where a high concentration of samples with residuals greater than 3RMSE are included in option 2) (see discussion section: AT inputs from rivers). On the other hand, the scattered samples could fit into option 1).

3RMSE is equal to 24.6 µmol kg$^{-1}$. Therefore, removing the samples beyond this threshold from the error computation should not give an error of less than 3 µmol kg$^{-1}$ as the reviewer suggests in the comment.

As neural networks try to fit a function with the provided data in the training set, removing the samples that do not fit well to the function created by the network is equal to removing their "bad" influence in this function. That is, in some way the network also tries to model the variability due to both non-included inputs or data with uncertainties. Therefore, the network trained without these samples could reach a more accurate function by not having samples that would need some other variable as input or uncertainties in any of the variables measured.

As you can see in Table 2 and Table 3, the RMSE is lower in almost all areas if NNw3RMSE is used to compute AT. Furthermore, the independent test in the time-series shows the same results (although they are not included in the MS). Therefore, based on these results, we offer the two networks to the scientific community to consider choosing one or the other according to the area of the study. Finally, we chose the NN to build the climatology to include the Arctic with the least possible error.

We have changed this paragraph to clarify this step.

**AC**: "As a last step we eliminated the data points with a difference between measured and computed $A_T$ with the selected network (residuals) beyond ±3RMSE and then retrained the network as above. This procedure was used to identify regions where the network was unable to obtain accurate values and to improve the network mapping in the other areas omitting in this way data that the network could be trying to model without having the appropriate input variables or because they could be data with high measurement errors. Although a well-trained neural network avoids modeling the error, high errors could slightly modify the derived function in a negative manner. The network derived from this process will be referred as NN±3RMSE."

**RC**: L171: What does it mean to "illuminate the complexity" of a neural network? Be more specific.

**AR**: The meaning is explained after the comma: "to determine the contribution of each predictor variable in the output". Furthermore, is quite similar to the title of the referenced paper by Olden and Jackson (2002): "Illumination the black box…".

**RC**: L208: Random division of the datasets is inadequate for your test. See main points.

**AR**: Random division is meant to have more relations between input variables and $A_T$ in the training set (greater coverage in the 4 dimensions: latitude, longitude, depth and

time). Therefore, the network is trying to model all the possible variability in the GLODAPv2 data and a random test set probably includes all this variability to evaluate it. In the new test that we have added to the section of the subsurface layer hypothesis deleting all the winter data in all the ocean you can see how the error in this deleted set is even lower than in the initial test set (6.8 vs 8.5 µmol kg$^{-1}$). This test also contributes to the reviewer's suggestion to leave a complete area without data to test the network there and the error is similar to that of NN. Also see the answer in the main points.

**RC**: L210: How does "The samples with residuals beyond _3RMSE are 1% of the global dataset: : :" align with "99% of the GLODAPv2 dataset used was modelled by the network with a root-mean-squared error (RMSE) of 5.1 _mol kg-1." I'm guessing this is referring to the 2nd NN, but I was confused for a long time before this statement started to make sense.

**AR**: "global dataset" is the same as "GLODAPv2 dataset used". The neural network is the first one: NN. It has been changed.

**AC**: "The samples with residuals beyond ±3RMSE are 1% of the global dataset" → "The samples with residuals beyond ±3RMSE are 1% of the GLODAPv2 dataset"

"99% of the GLODAPv2 dataset used was modelled by the network…" → "99% of the GLODAPv2 dataset used was modelled by the NNGv2…"

**RC**: L228: This sentence doesn't make sense to me… I suspect "disengage" is an incorrect word, but I'm not sure.

**AR**: Changed.

**AC**: "disengage" → "alter"

**RC**: L235: It's not always riverine TA that is the problem… often it is rivers with little or no TA that dilute seawater TA in a way that is distinct from the mixing patterns in the open ocean.

**AR**: The dilution of AT because of rivers with little or no AT should not be a problem. The dilution process is the same that in the precipitation process in open ocean and this is not a problem at all to the network.

**RC**: L235: this paragraph presents a weak argument against removing the region… fortunately, the argument for omitting that region was not made… omit this text, but instead estimate uncertainty regionally more quantitatively.

**AR**: 77% of the GLODAPv2 samples are well modeled in the Beaufort Sea and 91% in the North Sea. These high percentages suggest including the areas in the climatology is a good consideration. The regional error estimation is in Table 3.

**RC**: L251: "The zones defined.." this sentence is vague.

**AR**: In this paragraph we are discussing the results in Table 3, that is, AT computed with Takahashi et al. (2014) relations and AT computed with our neural networks. Therefore, the zones are those defined there. We have changed the sentence for a better understanding.

**AC**: "The $A_T$ computed in the zones defined in the Arctic have higher RMSEs in the two approaches (Takahashi et al. (2014) and this study; Table 3)"

**RC**: L252: considerably

**AR**: Changed.

**RC**: L261: what is meant by this sentence? Clarify.

**AR**: Except in the Artic, NN3wRMSE computes AT with a lower error than NN in almost all the areas defined in Table 2 and Table 3. The sentence has been changed for a better understanding.

**AC**: "Although NN±3RMSE computes AT with lower errors than NNGv2 in the non-Arctic areas, in a global view the improvement is relatively small (Weighted RMSE in Table 2 and Table 3)"

**RC**: l269: what is meant by this sentence? Clarify.

**AR**: The formation and degradation of organic matter is reflected in oxygen and nutrients. We have changed the sentence by this one.

**AC**: "The formation and degradation of organic matter is reflected through both oxygen and nutrients variations."

**RC**: L273: S and T collectively provide information regarding interior ocean density and mixing patterns, which are important for predicting TA: : : it is not clear what you are suggesting about the link between T and CaCO3.

**AR**: We are suggesting that the influence of temperature in both CaCO3 and organic matter cycles could make this variable a good proxy to capture the AT variability because of these processes by the network. We have referred the study by Lee et al. (2006) where this is said.

**RC**: L278: "The bias is relatively low in the three time-series with the highest number of data. The AT computed by the NN at KNOT and K2 is higher than the measured one.

Summed to the previous test, this independent test with a seasonal time resolution shows the good generalization of the NN." The first two sentences here are difficult to understand and the last one does not make sense to me.

**AR**: This paragraph describes the statistics in Table 4 to show the good behavior of the network in modeling independent data. The first referred sentence is describing the good performance in most of the data used as test here (HOT, BATS and ESTOC). The second one is to reflect that in both KNOT and K2 the network performance is not as good as in the other time-series. We have changed the text for a better understanding.

**AC**: "The bias is relatively low in the three time-series with the highest number of data (HOT, BATS and ESTOC). The $A_T$ computed by the NNGv2 at KNOT and K2 is slightly higher than the measured one, probably because of the influence in the $A_T$ variability of some variable not included as an input of the network (although an offset in the measurements of any of the inputs could also give this result). Summed to the previous test, the statistics obtained in this independent test with a good seasonal time resolution shows the good generalization of the NNGv2."

**RC**: L286: This logic does not follow.

**AR**: This paragraph has been deleted and a new one was added including the discussion about $A_T$ computation using the code of both LIARv2 and CANYONB. See in the new version of the manuscript attached at the end of the comments.

**RC:** L302: This sentence does not seem to fit with the rest of the paragraph.

**AR**: This part has been changed. See in the new version of the manuscript attached at the end of the comments.

**RC**: L310-L313: I don't understand these sentences.

**AR**: The properties analyzed in the four areas defined in Fig. 2 show, as expected, a higher monthly variability in the ocean surface than in the subsurface layers. The intra-annual variability depicted in Fig. 7 is likely to be typical also of larger regions within each ocean basin. The surface conditions in winter of the analyzed properties are quite similar to those in the subsurface layer in four consecutive months to winter in all ocean basins (Fig. 7). Thus, using the subsurface layer allows to retrieve winter surface conditions in other seasons. See also in the new version.

**RC**: L310: intra-annual would be clearer as "seasonal"

**AR**: Changed.

**RC**: L340-342: I don't understand the logic.

**AR**: Changed.

**AC**: "This result shows the importance of the inclusion of other predictors besides salinity in the network and the non-linearity of the method proposed in this study to explain nearly all the $A_T$ variability"

**RC**: L354: Isn't that figure 10?

**AR**: Thanks, it is. Changed.

**RC**: L363: Why a change of grid for a time series?

**AR**: There is no time-series discussion in this section. Here, we are comparing climatologies. The comparison referred to in L363 is between the climatology of Takahashi et al. (2014) and ours. Their climatology is in a 5°x4° grid and ours in a 1°x1° one. Therefore, we averaged our climatology to their grid which in some way leads to an error component in the comparison between the climatologies. Finally, the sentence has been changed.

**AC**: "Furthermore, the transfer of our climatology to the coarser grid of Takahashi et al. (2014) for the comparisons may enhance dissimilarities."

**RC**: L380: what is meant by a "continuity" in the differences?

**AR**: The largest differences in the referred comparison are quite patchy except in the two referred zones. We have changed the sentence for a better understanding.

**AC**: "there is a large continuity in the differences" → "there is a large spatial continuity in the differences"

**RC**: L393: I couldn't get the code to work: : : some info: : : ăˇA ´c Tried with the directory with NN as the active directory o As well as just with the directory with NN on the Matlab path ăˇA ´c Tried entering within a script and on the command line ăˇA ´c Tried with Matlab r2014b with the NN toolbox (also 2018b, but without the NN toolbox) ăˇA ´c With inputs as single or double precision numbers ăˇA ´c Entered: o AT_values=Neural_network_object(data_inputs); o AT_values=NN(data_inputs); o AT_values=NN_w3RMSE(data_inputs); Invariably got the response: "Undefined function 'NNw3RMSE'/'NN'/'Neural_network_object' for input arguments of type 'single'/' double'." I recommend making the instructions a bit more clear. I also recommend adding an example calculation so users can be sure they are getting the expected answers. You'll know you are done when the coauthors can use the function reliably without additional guidelines.

**AR**: The code is only one line. Only the neural network object and the input matrix in the form explained in the readme.txt file are needed. We have attached a matrix of

random inputs from GLODAPv2 and a very simple script. It has been successfully used by other colleagues. We have added this material on the same site as previously.

**RC**: Figure 7 and elsewhere: Uppercase theta should be reserved for conservative temperature… use lowercase.

**AR**: Changed.

**RC**: Figure 8 and elsewhere: The font is too small.

**AR**: Changed.

**RC**: Figure 10, right panel: How is the RMSE smaller than the bias? Clarify what is being shown here if there is a good reason.

**AR**: The measured seasonal variability is well represented by the climatology (low RMSE) but there is an offset between the values. The offset is because of the difference in the data inputs (monthly average of the measured data in the time-series vs. WOA13). Passing the measured data through the network, the bias is lower and positive (0.9 $\mu$mol kg$^{-1}$). This fact demonstrates that the network is not the cause of the relatively high bias of the other mentioned comparison. Therefore, Figure 10 was made for showing the good performance of the network and the differences in the climatological data from WOA13 and those obtained from averaging the measured data at time-series locations (the latter of which are the only locations where the neural network climatology can be properly compared with climatological measured data).

[revised manuscript text omitted]

---

## Author Comment (AC3) · 26 Feb 2019

**RC**: Referee comment; **AR**: author's response; **AC**: author's changes in manuscript.

**Referee #3 response**

**RC**: The authors describe a neural network approach to derive an algorithm to estimate AT from concurrent Lat/Lon, depth, T/S, oxygen and nutrient (nitrate, silicate and phosphate) inputs, based on GLODAPv2 data. They use this approach with monthly climatological fields from WOA13 to establish a global, depth-resolved, monthly AT climatology. The manuscript is clearly-structured and well-written.

I see three critical points, (1) the neural network topology selection and the second round of neural network training without control for overfitting, (2) the adequate representation of (surface) seasonality in the training data, by the neural networks, and the derived monthly climatology, and (3) the placement and comparison with other recent work on AT estimation based on GLODAPv2-trained algorithms. I therefore suggest major revisions to the manuscript.

**AR**: Thank you so much for the thorough revision of the manuscript. We hope to improve it based on all the comments of the 3 referees. At the end of the answers is attached the new version of the manuscript for a global view.

**RC**: Major points:

(1a) The authors describe training of their neural networks in general terms, however, some important details remain missing.

• The selection of the best performing neural network appears subjective and is not made transparent. This needs to be improved. E.g., l.161: What criterion has been used to assess "best generalization in the initial testing dataset"?; l. 204f: 128 neurons kind of fall from the sky. Figure S1 would probably be more instructive to show RMSE for training and testing set vs. number of neurons, to make the authors' reasoning more transparent.

**AR**: The selection of the best neural network is based on the RMSE in the test set. We have added it in L.161, although this whole paragraph has been modified also considering the comments of one of the other referees.

The selection of the optimal number of neurons is based on the explanation given in L-142-150. However, we have added the suggested Figure S1.

**AC**: "The training procedure was carried out in MATLAB. We tested 16, 32, 64, 128 and 264 neurons in the hidden layer based on the results of Velo et al. (2013). For each number of neurons, we trained 10 networks always using the same 90% of GLODAPv2 for training (Fig. 2, Static level). The remaining 10% was used as a static test (Fig. 2, Static level). Both subsets contained samples randomly distributed in the ocean to evaluate the maximum possible relationships between the input variables and AT through all oceanographic regimes, that is, to capture most of the variability in all the

variables and not restricting the sets to specific areas. Each of the 10 networks starts the training procedure with random weight and bias values and a random division of the training static dataset into three portions: 70% for training, 15% for testing and 15% for validation (Fig. 2, Dynamic level). These differences make minimization of the cost function different for each network due to the complexity of the weight-error space and, consequently, their different starting points in that space. As each network is different, keeping static sets allows one to determine which network best generalizes in the same test set. The selected network is the one that produces the lowest RMSE in the training data (validation + training dynamic) and in the test data (static + dynamic), considering a non-significant difference between both RMSEs to prevent overfitting. The network derived from this process will be referred as NNGv2.

[Figure]

**Figure 2. Division of the data for the training of the network. The data in the sets of the static level is the same for all the networks to train. The data in the sets of the dynamic level is randomly selected for each network to train.**

[Figure]

**Figure S1. RMSE variation with the number of neurons of the network for the lm algorithm. Training data contain the dynamic training set and the dynamic validation set, that is, 76.5% of the GLODAPv2 dataset used in this study. Testing**

**data contain both the static and the dynamic test sets, that is, 23.5% of the GLODAPv2 dataset used in this study."**

**RC**: • Do the authors use weight regularization of the network weights? I presume so, at least for the Levenberg-Marquardt variants but probably also for their Bayesian regularization. This should be stated. It should be stated as well how the regularization (hyper-)parameter/weight was chosen (i.e., the balance between data accuracy/loss and weight penalty/loss terms in the cost function; or in other words the balance between accuracy and generalization behavior within the given network topology).

• What exactly is meant by Bayesian Regularization (l. 141 with reference to MacKay, 1992)? Please be more explicit here. If you used a certain, e.g., Matlab implementation/toolbox, make reference to it. MacKay (1992) describes at least three levels of Bayesian treatment, from (I) finding the 'best' (most-probable) set of weight parameters including their regularization (i.e., preserving generalization behavior by avoiding too specialized weight distributions) through (II) finding the 'best' hyperparameter values (i.e., objectively assigning the balance between data loss and complexity/regularization) to (III) model comparison (e.g., quantitatively rank different models or neural network topologies). It seems to me that only (I) has been used here? Please clarify. Also note that, if implemented correctly (!), Bayesian regularization doesn't need cross-validation like, e.g., a backpropagation Levenberg-Marquardt learning scheme.

**AR**: We have added a more extensive description of the two techniques used in this study and referred to the MATLAB toolbox used, as the reviewer suggested. However, we did not add a full description explaining the math and all the processes behind the different algorithms inside the neural networks. We have added the necessary references for the readers who are looking for more detail. The scope of this manuscript is to explain the basics to create a basis for the readers to understand how we achieve the objectives.

**AC**: "Two different training techniques were tested: the Levenberg-Marquardt method (lm) and the Bayesian Regularization (br) (both detailed in Hagan et al., 2014). In a similar study, Velo et al. (2013) demonstrated that these techniques give the best network performance among those they tested. Except for the number of neurons, the two algorithms were implemented with the default options of the MATLAB functions trainlm and trainbr (detailed in Beale et al., 2017). These two functions prevent overfitting in different ways. The trainlm function usually needs to be fed with the data divided in three sets: a training set to obtain the relationships between variables, a validation set to prevent overfitting and a test set to compare different networks. Here, the training was stopped when the error in the validation set increased during 6 consecutive iterations of the training process to avoid overfitting. This process is known as early stopping (Hagan et al., 2014). The final values of the network weights and biases are those reached before the first of these iterations. The trainbr function adds a regularization parameter to the cost function to make the fit smoother in order to avoid overfitting. The validation set is not present in this technique. The end of the training is

based on network convergence through parameter stabilization by an automatic process known as automated Bayesian Regularization (Hagan et al., 2014; Beale et al., 2017). See Beale et al. (2017) and references therein for a detailed description of the two functions tested."

**RC**: (1b) I think the two-step training of the networks with elimination of the testing data must be avoided (with a backpropagation/LM algorithm). Optimization of the network's parameters doesn't stop after training with the 70/15/15 % training/validation/testing data set. It continues well throughout the 80/20/0 % step, where the authors no longer have control over or means to assess overfitting. The authors' conclusion (l.165) is invalid. Given that, e.g., "[the authors] find no improvement by increasing the amount of data points in the training set" (l. 207), I don't see the point in making this questionable second step. Instead, this re-optimization of weights without control for overfitting makes the method vulnerable. It should be removed thus closing this open flank without loss in performance.

**AR**: It was a mistake. We deleted the set which is used to validate the model (named as test set in MATLAB sets division). We have changed this paragraph. The validation set which is used to prevent the overfitting is maintained. The static test set (named in this way in the new version of the manuscript) is maintained to choose the best network. We deleted the dynamic test set. We make this step to test if to have more data to train a neural network would improve the fitting. The results suggest that no more data is necessary and maybe include other input variables would improve the fitting.

**AC**: "Once we found an adequate network configuration, we increased the amount of data in the training dynamic set to capture more relations between the inputs and AT. The new percentages of the dynamic sets were: 80% training, 20% validation and 0% testing. The latter set is only necessary to compare different models and is not used during the training. However, the static test set was held to evaluate the generalization of each of the 10 networks to select the best one."

**RC**: I do see the NNw3RMSE run critical, too. In essence, the authors level out areas of the ocean with higher-than-average variability (>3* global-mean-RMSE *samples* are removed, i.e., only the subset of samples that fit to the mean in these areas is retained). They do this to "improve the network mapping in the other areas" (l. 169). This *spatial* difference/distinction should be captured by the sampling position input (Lat, sLon, cLon, Depth), shouldn't it? I would argue that the (small) improvements they see in certain subregions between the NN and the NNw3RMSE runs is only due to a different local minimum found during neural network training of the one neural network selected for NN vs. the one neural network selected for NNw3RMSE, and not thanks to the omission of data in an at most adjacent or even unrelated ocean region (e.g., Equatorial Upwelling Pacific, while most samples >3*global-mean-RMSE are found at high latitudes/North Sea). Again, given that "the difference in the weighted RMSE of the two networks [NN and NNw3RMSE] is not significant" (l. 247; l.254) and that the

authors consider NN the best candidate for users (l. 263), I'd suggest to drop the NNw3RMSE network.

**AR**: Most of the data deleted for the retrained network (NNw3RMSE) are from the Beaufort Sea. In Table 3 you can see how big the difference is between the errors computed by the two networks. This result suggests that the NN is obviously trying to fit the data of this area. In this case, it is clear that omitting certain data causes a large difference between the networks. Although the improvements are not as high in many of the areas defined in Table 3, these improvements in almost all of the zones suggest that they are because of this data deletion instead than the different local minimum reached in the error function. The NN training tries to fit these data in some way and it could be fitting a function not good enough to fit the data of the areas that are easier to model as the NNw3RMSE does. However, the suggestion that the reviewer makes could also be a possibility and thus we have included it in the discussion.

We decided to keep the NNw3RMSE based on the suggestion made previously about the reader's ability to choose one or another network regardless of the reason for improvement in certain areas. Moreover, it has been checked in a new application designed for QC (https://github.com/ocean-data-qc/ocean-data-qc) as this network computes better AT in recent data not included in GLODAPv2 than NN or CANYON-B.

**AC**: "As a last step we eliminated the data points with a difference between measured and computed $A_T$ with the selected network (residuals) beyond ±3RMSE and then retrained the network as above. This procedure was used to identify regions where the network was unable to obtain accurate values and to improve the network mapping in the other areas omitting in this way data that the network could be trying to model without having the appropriate input variables or because they could be data with high measurement errors. Although a well-trained neural network avoids modeling the error, high errors could slightly modify the derived function in a negative manner. The network derived from this process will be referred as NN±3RMSE."

**RC**: (2) I think it is courageous to derive a monthly-resolved product from GLODAPv2 data, which in many ocean regions is far from being monthly yet seasonally-resolved. This needs further elaboration and the seasonal character needs to be demonstrated clearly.

To tackle the scarcity of winter time observations, the authors state that the lack of surface information during winter can be circumvented by using spring time observations of subsurface waters that retain the winter water signature, illustrated by figure 7. (l. 187-194).

Fair enough, but this information doesn't tell the neural network to learn it that way nor does it imply by any means that the neural network recognizes this connection. Even if winter water properties are similar between spring subsurface and winter surface samples (as in the climatological WOA13 data of figure 7), the vertical sampling location (Depth) is still different, thus ending up in a different area of the neural network input data space - giving potentially very different AT output.

The first step to convince me of this 'seasonal winter gap filling' would be to add the predicted surface and subsurface AT to figure 7 - which should approach each other during winter like the water properties.

A second step would be to give better quantification of the seasonal cycle where possible. This is probably limited to the time series stations and the North Atlantic. If the training data are seasonally well-resolved and the neural network training picks up this seasonality adequately, the seasonal cycle's amplitude from the NN (and measured inputs) should be of the same magnitude as the observed seasonal cycle's amplitude. If the training data do not reflect full seasonality, the NN tend to underestimate the seasonal cycle - with a flat line as the extreme. Such a comparison should complement the for now only qualitative assessments (e.g., figure 5).

Moreover, the (sub-)polar North Atlantic should be added as region for the sub-surface hypothesis due to the high interest in the carbon cycle in this area.

**AR**: To show the potential of the neural network to obtain accurate values in winter times we have added a new test in the subsurface layer hypothesis section. We trained a new network without the winter data in the same way as the original network. If the network is able to capture the winter relations between inputs and AT from other seasons, the excluded winter dataset from the training should be model by the network. The good statistics show how the network obtains the winter relations for all the ocean areas where the winter data was measured in our GLODAPv2 dataset without the need to be trained with winter samples.

In a very complex error space, with more than 1000 dimensions like the one resulted from our NN, it is very difficult to know how any input variable is related with the others and with the output variable. Therefore, "depth" (and the other input variables) may not be assimilated by the network in a way that we may understand. What is clear with this new test is that the network is able to model winter data even if it does not have such winter data for training.

The subpolar North Atlantic has been evaluated by Vázquez-Rodríguez et al. (2012) as we stated in the manuscript and therefore, we evaluated other zones within the WOA13 data. Nevertheless, this new test has assessed an extensive coverage of the ocean. It can be seen in the new version of the manuscript attached at the end of the answers.

About the North Atlantic time-series that the reviewer refers to (we suppose Irminger and Iceland), we could not include them in the study because no AT was measured there. However, we have evaluated the network performance in these regions with AT computed from pCO2 and DIC (although the temperature at which pCO2 was measured is not clear in the dataset we downloaded from CDIAC) and the statistics are similar to those of the time-series included in this study, with a seasonal cycle amplitude similar to that of the computed AT from measured pCO2 and DIC.

**RC**: (3) Since the publication of the GLODAPv2 data set, there have been other works that use the data compilation to establish algorithms for AT estimation. Two of them are mentioned (LIARv1, Carter et al. 2016, and CANYON, Sauzede et al. 2017), however,

the manuscript falls short on setting their own work into perspective of the state-of-the art published literature.

(a) Both methods mentioned have received updates (LIRv2, Carter et al. 2018, and CANYON-B, Bittig et al. 2018), to which the comparison of the present work should be made. Both updated algorithms are publicly available as Matlab code and use overlapping (but fewer) inputs as the authors' approach, i.e., there is no obstacle to apply them to any of the authors' data.

(b) The authors already do a decent job in assessing their work with surface-only climatologies (e.g., Lee et al. 2006), but the authors need to demonstrate more clearly how the present work improves / compares with existing, global, depth-resolved algorithms of AT (e.g., see above).

E.g., in terms of accuracy on all their time series data, not just HOT (section 3.2), surface seasonal amplitude (see point 2 above), complexity in terms of input data requirements, etc.

Interestingly, the authors don't use the year day as input either (same as LIR and CANYON-B), and nonetheless get good surface seasonality.

This point (3) is important to improve, since it will give the authors the argument of why use their algorithm (or one of the others) to derive an AT climatology from WOA13 fields, which is the main subject of this work (following the title).

**AR**: We have added a general comparison with the two updates of the referred methods. We had only focused on the algorithms of Lee et al. (2006) and Takahashi et al. (2014) since they were created to generate a monthly climatology, as in this study.

First, we showed in our GLODAPv2 dataset how our NN fits the AT better than LIARv2 and CANYON-B. Furthermore, we compute the statistics excluding the data were the QC was not made since both referred methods did it to train their algorithms.

**AC**: In 2. Methodology: "The recent methods to compute $A_T$ proposed by Carter et al. (2018) and Bittig et al. (2018) (LIARv2 and CANYON-B respectively) were also compared to the one proposed here. LIARv2 is based on multilinear regressions (MLRs) including the same predictors used in the present study, excluding phosphate (sample position, salinity (S), potential temperature ($\theta$), nitrate (N), apparent oxygen utilization (AOU) and silicate (Si)). This method is composed of 16 equations with a different combination of the input variables, always maintaining the salinity input in each one. The computations with LIARv2 were obtained by the equation with the lowest uncertainty estimate in each sample that this method determines (Carter et al., 2018). CANYON-B is based on a Bayesian neural network derived from GLODAPv2 data including position, time, salinity, temperature and dissolved oxygen as predictors. The two methods were applied on the GLODAPv2 dataset used here and the on a subset excluding the samples where the quality control (QC) of $A_T$ was not done (QC procedures detailed in Olsen et al. (2016) and references therein)."

In 3.1 Neural network analysis "The newest methods in the AT computation (LIARv2: Carter et al., 2018; CANYON-B: Bittig et al., 2018) model the GLODAPv2 AT with higher errors than the NNGv2 (Table 4). An analysis in a GLODAPv2 subset excluding

the samples where the 2nd (Olsen et al., 2016) QC was not done for AT shows a reduction of the error in these three methods, being CANYON and NNGv2 the lowest (Table 4). All the equations are used to compute AT in the GLODAPv2 dataset when the computation is allowed to be made by the equation with the lowest uncertainty in each sample (Carter et al., 2018). The most used equations are 10 (S, N, Si), 15 (S, AOU) and 14 (S, N), which are used in about 50% of the samples. The equation that used all the input variables (1) is only used to model 3% of the GLODAPv2 samples. Surprisingly, when only this equation is used to compute AT in GLODAPv2 dataset, the error is lower than those obtained with the free election of the equation based on the lowest uncertainty. That result shows the potential of include all possible inputs related with the AT variability, although reasonable results can also be reached with the equations that do not use all the input variables. CANYON-B is an example of using relatively few input variables (position, time, temperature, salinity and oxygen) and getting good results (Table 4). Probably, the non-linear character of the neural networks, like the one used in CANYON-B, gives the high potential to this kind of methods to fit complex functions even with few input variables. However, the NNGv2 designed in the present study is the best option to model more GLODAPv2 data better than the other methods (lower RMSE) and therefore to use the mapped inputs-output relation in order to create the monthly climatology. The availability of all the variables used as inputs of the NNGv2 in WOA13 also contributes to make this method the best choice. Furthermore, methods like CANYON-B which include a predictor that explicitly accounts for the time variation of AT (decimal year in the case of CANYON-B), are not suitable to build a monthly climatology since they generate an unrealistic seasonal amplitude, at least at high depths. This has been checked used WOA13 monthly climatologies (temperature, salinity and dissolved oxygen) as inputs of CANYON-B to compute AT at different depth layers. As an example, in the 3000m depth layer, seasonal amplitudes up to 40 $\mu$mol kg-1 were obtained in large areas mainly located between 30 and 60ºS.”

In 3.2 Time-series validation “The LIARv2 and CANYON-B methods to compute AT also model the time-series data quite well (Table 6). Significant differences among the three methods are obtained in HOT and ESTOC. In HOT, NNGv2 and CANYON-B reach a better fit of AT than LIARv2 suggesting that a non-linear technique is more adequately to model AT in this area (Table 6). In ESTOC, NNGv2 and LIARv2 are the best options to model the AT variability (Table 6). Here, the AT computed with LIARv2 with the option of the free equation choice activated results in a greater election of the equations that include nutrients as predictors. This result show how in this area the inclusion of nutrients as predictors contributes to improve the model of AT. Like NNGv2, both methods have a considerable bias in K2 and KNOT (Table 6) that reinforce the two reasons suggested previously.”

**Table 4. RMSE and bias obtained with NN, LIARv2 (Carter et al., 2018) and CANYON-B (Bittig et al., 2018) in both GLODAPv2 dataset and GLODAPv2 dataset without the samples where AT QC was not done.**

| Approach | RMSE ($\mu$mol kg$^{-1}$) | bias ($\mu$mol kg$^{-1}$) | n |
|---|---|---|---|
| NN_GLODAPv2 | 8.2 | 0.02 | 246221 |

| | | | |
|---|---|---|---|
| LIARv2_GLODAPv2 | 11.4 | 0.08 | 246221 |
| CANYON-B_GLODAPv2 | 11.4 | -2.8 | 246221 |
| NN_GLODAPv2_onlyQC | 6.6 | 0.06 | 215332 |
| LIARv2_GLODAPv2_onlyQC | 8.2 | 0.06 | 215332 |
| CANYON-B_GLODAPv2_onlyQC | 7.8 | -3.2 | 215332 |

Table 6: RMSE and bias between measured $A_T$ and the $A_T$ computed with both LIARv2 and CANYON-B methods. The comparison was done for the same samples evaluated in Table 5.

| | LIARv2 | | CANYON-B | |
|---|---|---|---|---|
| | | bias | | bias |
| Time-Series | RMSE ($\mu$mol kg$^{-1}$) | ($\mu$mol kg$^{-1}$) | RMSE ($\mu$mol kg$^{-1}$) | ($\mu$mol kg$^{-1}$) |
| HOT | **6.6** | **-0.6** | **5.8** | **-0.6** |
| BATS | **6.3** | **0.1** | **6** | **-0.4** |
| ESTOC | **3.4** | **0.8** | **4.2** | **3.2** |
| KNOT | **4.8** | **-6.6** | **4.5** | **-7.2** |
| K2 | **3** | **-3.0** | **3** | **-3.3** |

Minor points:

**RC**: l. 60: remove oxygen. Nutrient changes contribute to a change in AT, oxygen itself does not contribute to the charge/acid/base balance.

**AR**: Changed

**AC**: "(Millero et al., 1998; Fine et al., 2017). Organic matter cycling can also contribute to AT changes. This mechanism can be reflected through the consumption and regeneration of nutrients and oxygen (Brewer and Goldman, 1976; Wolf-Gladrow et al., 2007)."

**RC**: l. 105: The number of neurons in the output layer is adjustable? It seems to be n=1 for just AT, isn't it?

**AR**: Changed

**AC**: "The hidden and output layers are composed of neurons. The number of these elements in the hidden layers is adjustable and in the output layer is dependent on the number of network outputs. The neurons are formed by a series of weights, a bias, a summation, and a transfer function (Russell and Norvig, 2010). They are the connections between the layers."

**RC**: l. 124: "as previously described" - not yet done, remove.

**AR**: Removed.

**RC**: l. 138 and 139: Which spurious oxygen value was removed / Where can it be found? (To allow reproduction by others.); Name the ocean time-series or give their GLODAPv2 cruise IDs.

**AR**: Added cruise, station and bottle of the sample with spurious oxygen value. Second sentence has been deleted and it has been commented in L. 182 (in the first version of the manuscript).

**AC**: "From these, we removed one record due to its spurious oxygen value (O2=1026.9 µmol kg$^{-1}$ cruise=102; station=4; bottle=5)."

**RC**: l. 238: "As an argument ... areas." Unclear.

**AR**: Changed.

**RC**: l. 273: Depth is rather associated as vertical sampling position.

**AR**: Of course, it is. But, together with temperature and salinity it can help to model the variability of AT because of CaCO3 cycle.

**RC**: l. 279: Any ideas why there is such a bias? Should be commented.

**AR**: Added.

[revised manuscript text omitted]

---

## Referee Report (RR1)

**Review of 1st manuscript revision of "A global monthly climatology of total alkalinity: a neural network approach" by D. Broullón for ESSD**

**April 5, 2019**

Thank you for revising and trying to improve your manuscript. The authors better explain their network training procedure and what dataset they use for which statistics. They also make better comparisons of their method to other state of the art approaches (though there is still room to make them adequate, see below).

However, the manuscript did not improve sufficiently. There are still significant shortcomings, which require major revisions.

Based on the initial submission, their approach showed good potential for a fully seasonal, monthly climatology, but the authors fail to clear the concerns raised by all three reviewers in this revision. This encompasses both the seasonality of their NNGv2 network based on GLODAPv2 training data (I am afraid their 'reinforcement' of the subsurface layer hypothesis does not reinforce anything in its present form.), as well as the seasonality of the produced $A_T$ climatology from WOA13. It encompasses as well the need for a more robust uncertainty assessment of the produced $A_T$ climatology.

Find below my comments ordered from critical to major to minor.

**1 Critical points**

**1.1 Seasonality at depth**

There is a lack of transparency as to what depth the climatology of $A_T$ is seasonal or not.

l. 406 states that "seasonality disappears almost completely below 500 m depth; not surprising due to the lack of seasonal resolution in the climatologies of nutrients in WOA13 below this level." but otherwise, the authors claim to provide a "global monthly climatology of $A_T$ on 102 depth levels" (l. 223 and abstract l. 31), i.e., to full depth?

WOA13 input data have monthly resolution down to 500 m for the nutrients, and down to 1500 m for T, S, and $O_2$. WOA13 input data have quarterlyresolved files down to 500 m for the nutrients, and down to full 5500 m depth for T, S, and $O_2$. Finally, WOA13 has annual mean files down to full 5500 m depth for nutrients, T, S, and $O_2$.

The quarterly-resolved fields of T, S, and $O_2$ show in some parts strong differences at high depths, e.g., at 3000 m (see attached figure 1, left column). The authors appear to interprete that as seasonality ("The seasonal amplitude of $A_T$ is progressively reduced at depth" l. 405f/Figure S7, and l. 310-315)?

For perspective, WOA13 quarterly variations at 3000 m depth have (e.g., in the Southern Ocean) a range up to 0.4 °C (on a total range of variability of ca. 40 °C), 0.05 psu (on a total range of ca. 2 psu), and beyond 40 $\mu$mol/kg (on a total range of ca. 400 $\mu$mol/kg). This certainly exceeds any expectation for a seasonal cycle, e.g., of oxygen, and demonstrates rather a data coverage and/or mapping issue. (Please consult / check the data coverage fields 'x_dd'!)

This has three consequences:

1. The authors must decide until what depth they use which monthly-/ quarterly-/ annually-resolved WOA13 input fields, determining until what depth they can claim to provide a monthly-/ quarterly-/ annually-resolved $A_T$ climatology.

   This must be 500 m, if they decide that their seasonality is caused by the organic matter cycle, reflected through both oxygen and nutrients variations (l. 319/321) (summed 57 % relative importance). It may be 1500 m if they decide that *monthly*-resolved oxygen (16 % relative importance), together with *annual means* of the nutrients (summed 41 % relative importance), may provide a sufficient driver for $A_T$ seasonality to the NNGv2 network, but this already must be clearly justified. It would be quite a stretch for any reasonable seasonality below 1500 m, and I would suggest to revert to annual mean WOA13 fields, rather than the quarterly ones, to built the $A_T$ climatology. Please consult the 'x_dd' data coverage fields, too, to assess if sufficient data went into the monthly / quarterly fields for a robust seasonality – or a robust assessment at all.

2. The seasonality and its limits must be made transparent through the author's work/manuscript. It should not be the task of the reviewer / user to check.

3. The CANYON-B/WOA13 comparison (l. 310-315) must be moved to the Climatology section 3.4 rather than the Neural network analysis 3.1 and discussed accordingly. A computation method "using relatively few input variables (position, time, temperature, salinity and oxygen)" (l. 305) is more prone to bad input data in one variable than a method that uses all the variables as inputs (l. 309). Particularly, (1) if the only biogeochemical predictor may be biased (oxygen; CANYON-B) rather than just one out of a total of three (LIARv2) or four (NNGv2), of which the three nutrients nitrate, silicate, and phosphate just have a WOA13 *mean annual* field below 500 m (!); (2) if exactly this predictor has a "seasonal" $O_2$ amplitude

[Figure]

Figure 1: Left column: WOA13 "seasonal" amplitude at 3000 m of temperature (top), salinity (middle), and oxygen fields (bottom) from WOA13 quarterly-resolved files, which are (probably?) used as input to the $A_T$ climatology and for CANYON-B 'comparison' (l. 310-315). Right column: $A_T$ "seasonal" amplitude at 3000 m for the author's *monthly* climatology (probably?) based on WOA13 *quarterly* fields of T, S, $O_2$ and *annual mean* fields of nutrients as input (top), LIARv2 using WOA13 *quarterly* fields of T, S, $O_2$ and *annual mean* fields of nutrients as input (middle), and CANYON-B using just WOA13 *quarterly* fields of T, S, and $O_2$ (bottom). Note the correspondance of elevated patches of "seasonal" amplitude between WOA13 quarterly $O_2$ and CANYON-B $A_T$ in the Southern Ocean (a.k.a. 'garbage in, garbage out').

at 3000 m of up to 40 $\mu$mol/kg (versus the time / decimal year with a pretty modest variability of ca. 0.5 years on a total range of 40 years!); and (3) considering the strong correspondence at 3000 m between "seasonal" oxygen and "seasonal" $A_T$ amplitude (attached figure 1 right column).

To claim / blame the time for these variations in $A_T$ is pretty bold and wrong.[1]

Please correct the text, if you decide to keep it, and please make an effort for a balanced comparison of methods!

Please also correct the conclusion (l. 471/472), which neglects the impact of the quality of the WOA13 seasonality.

**1.2  Uncertainty of climatology**

Why is the monthly $A_T$ climatology not compared to the measured GLODAPv2 $A_T$ data? This would give a much more robust assessment than just two time series sites (HOT and BATS), at which the WOA13 input data arguably should be on the better side of the spectrum of possibilities, i.e., underestimating the climatology's uncertainty.

I don't see why such a comparison should be limited to *locations* with repeated sampling (l. 415) and not extended to *times/months* with repeated sampling (read: basin-crossing cruises as in GLODAPv2).

At the end of the day, the $A_T$ climatology should represent both temporal and spatial variability within its resolution – What better dataset to assess temporal and spatial variability than the largest available one, GLODAPv2? This should also include spatial / regional differences in the uncertainty.

At least some assessment of $A_T$ climatology uncertainty must be given before the dataset is acceptable for publication.

**1.3  Subsurface layer hypothesis**

Quoting from the text, the "winter relationship between inputs and $A_T$ needed to produce an all-season surface climatology are mostly preserved in [the] subsurface layer." (l. 214).

However, the authors try to reinforce the hypothesis (1) using the nowinter network on all depths rather than just the surface, and (2) the NNGv2 network is never evaluated on the same data as the nowinter network. In fact, the numbers in table 7 might be nice to show, but don't give any indication about the validity of the subsurface layer hypothesis.
* * *
[1]In addition, year-to-year $A_T$ variability, i.e., decimal year +/-1.0, is almost negligible in CANYON-B $A_T$ with WOA13 input. If the "seasonal" $A_T$ variations were to be caused by the decimal year variable, the representation inside the neural network would have to be strongly oscillatory, which contradicts the principle of early stopping / regularization to produce smooth network representations (e.g., l. 134-140; Hagan et al., 2014).

The lower winter RMSE may just be related to less variability (at all depths?) during this one season compared to the three nowinter seasons. Many other reasons are plausible, too.

What the authors should do: (1) As the authors suggest and do in l. 370, control that the nowinter network is comparable to NNGv2 on the domain it is trained on by providing statistics for NNGv2 and nowinter network on the GLODAPv2_nowinter dataset (full depth and surface; can be moved to the supplementary if desired); (2) Provide statistics for NNGv2 and nowinter network on the GLODAPv2_winter dataset using ***only surface*** data (above the subsurface layer defined in lines 358-362). Only if they are comparable, or at least not exceedingly higher for nowinter over NNGv2, or not exceedingly higher than surface RMSEs in other seasons (e.g., GLODAPv2_nowinter dataset surface only), this would reinforce the subsurface layer hypothesis. (That is, the exclusion of the (scarce) winter data did not degrade the winter surface predictions (← nowinter network and NNGv2 network on GLODAPv2_winter surface data) thanks to the still present signature in the spring subsurface layer (← GLODAPv2_nowinter training data and full GLODAPv2 training data).)

Other than that, the subsurface layer hypothesis remains a hypothesis, which I'd doubt the NNGv2 to recognize and would suggest to remove.

Figure 8: Same question as before: Why are calculated NNGv2 $A_T$ values not shown? They should.

**2  Major points**

- Table 2/3: Why are LIARv2 and CANYON-B not added here? They should be added!

- l.295. "The newest methods in the $A_T$ computation (...) model the GLODAPv2 $A_T$ with higher errors than the NNGv2 (Table 4)." **This is because both LIARv2 and CANYON-B used only the GLODAPv2 $A_T$ subset for training where the 2nd QC was done, whereas our GLODAPv2 $A_T$ data for training included samples, too, where the 2nd QC was not done.** "An analysis in a GLODAPv2 subset excluding the samples where the 2nd QC was not done for $A_T$ shows a reduction of the error [...]".

  NNGv2 results are not comparing to independent data in the GLODAPv2_no_secondary_QC subset because of correlations within cruises and the random splitting of cruises between testing/training, whereas LIARv2 / CANYON-B truly haven't seen any of these data.

- Table 5 and 6 should be merged, such as table 4.

- I still find it hard to justify the NNGv2_3RMSE network. The only clue to this one is that you remove an area (Arctic Ocean) with higher-than-average variability and, naturally, get better statistics. If you remove the same area from the NNGv2 assessment, you get the same, better statistics, too (table 1)!

  There is still a lack of justification for the NNGv2_3RMSE, and no, I don't think that a few decimal places better RMSEs in a few out of the regions in table 3 justify the 3RMSE removal – you can get the same better performance here or there by having a closer look at the 10 NNGv2 networks you trained! Also, the conclusions from the authors response are not supporting their argument and are not substantial. ("[...] In this case, it is clear that omitting certain data causes a large difference between the networks." I don't see a large difference. If you insist, please use an appropriate test to verify significance; the "improvements in almost all of the zones suggest that they are because of this data deletion instead than the different local minimum reached in the error function." That's only what the authors want to see, I'd still see a different local minimum as more plausible. And no further evidence is given that this may not be the case.)

  Please improve the NNGv2 vs. NNGv2_3RMSE network aspect or remove either one of the two.

- What about the seasonal amplitude of $A_T$ at the time series sites of measured $A_T$ vs. NNGv2-measured inputs-based vs. NNGv2-WOA13-based?

- To be complete, the subpolar North Atlantic should still be added to the current manuscript as test region for the current methods, even if it was the object of a previous work (Vázquez-Rodríguez et al., 2012).

**3  Minor points**

- l. 316: "The NNGv2 seems to associate the $A_T$ variability to the predictor variables *in coherence with the processes that contribute* to it."

  So, does it? Please give evidence or remove/rephrase.

- Table S1: What does 'HS' mean?

- Table S2/S3: Column headings 'relative ... lat>60° N' should probably correspond?

- l. 30: missing subscript $A_T$

---

## Author Response (AR2)

Response to Referee #1

**RC**: The revised MS reads well and it resolved the question I raised especially about the "subsurface layer hypothesis". I have no more major concerns, but only a few small problems:

**AR**: Thank you very much for your second revision of the manuscript. We are very happy to see that you do not have more major concerns.

**RC**: 1. Lines 166-167, "The new percentages of the dynamic sets were: 80% training, 20% validation and 0% testing. The latter set is only necessary to compare different models and is not used during the training"

Do you mean the reported AT predication is based on 90% training, 10% validation and 10% testing? What do you mean by "compare different models"? I may miss your explanation in the MS.

**AR**: The dynamic testing set is used to test the 10 different neural networks trained for each number of neurons. However, we have changed the approach a bit in this new version and we have decided to remove the test done increasing the amount of data in the training set.

**RC**: 2. The captains of supplement tables are still below the tables.

**AR**: Thank you. Changed.

**RC**: 3. Line 241-242, "In this area, 6.5% of GLODAPv2 samples have residuals beyond ±3RMSE and 83.1% of these samples are from the upper 100m (Table S2)" . The value here still mismatch the number in Table S2.

**AR**: Thank you. Changed.

**RC**: 4. Line 241-245, "A monthly analysis in the previously indicated ranges shows that the largest number of samples with residuals beyond ±3RMSE are from the summer months." This because you simply have more data in summer months than in winter seasons, but the percentage in summer may be the opposite.

**AR**: It could be a reason, but it is not like that. If you compare a spring month (e.g, June) with a summer month (e.g., September) that both have the same amount of samples (around 900), you can see how the first month has 2% of the samples with a computed $A_T$ with an error beyond 3RMSE and the second one a 20%. Therefore, that is not the cause and probably is because of the increasement of river discharge with high $A_T$ concentrations.

**RC**: 5. Line 277, Except for the zone with the lowest number of samples (Red Sea) and Okahotsk Sea

**AR**: Changed

**RC**: 6. Line 279, The AT computed in the zones defined in the Arctic have higher RMSEs in both approaches.

**AR**: Changed.

**RC**: 7. Can you merge Table 5 and 6 together?

**AC**: Merged.

**Response to referee #2**

**RC**: Thank you for revising and trying to improve your manuscript. The authors better explain their network training procedure and what dataset they use for which statistics. They also make better comparisons of their method to other state of the art approaches (though there is still room to make them adequate, see below).

However, the manuscript did not improve sufficiently. There are still significant shortcomings, which require major revisions.

Based on the initial submission, their approach showed good potential for a fully seasonal, monthly climatology, but the authors fail to clear the concerns raised by all three reviewers in this revision. This encompasses both the seasonality of their NNGv2 network based on GLODAPv2 training data (I am afraid their 'reinforcement' of the subsurface layer hypothesis does not reinforce anything in its present form.), as well as the seasonality of the produced $A_T$ climatology from WOA13. It encompasses as well the need for a more robust uncertainty assessment of the produced $A_T$ climatology.

Find below my comments ordered from critical to major to minor.

**AR**: Thank you very much for revising the new version of the manuscript. We have tried to improve it in this new version maintaining a balance in the modifications because of the favorable evaluation of the other reviewer, since his concerns have been solved in the previous version. Note that we have added the new MS below the answers with the highlighted changes so you can easily locate them.

**1 Critical points**

**1.1 Seasonality at depth**

**RC**: There is a lack of transparency as to what depth the climatology of $A_T$ is seasonal or not.
l. 406 states that "seasonality disappears almost completely below 500 m depth; not surprising due to the lack of seasonal resolution in the climatologies of nutrients in WOA13 below this level." but otherwise, the authors claim to provide a "global monthly climatology of $A_T$ on 102 depth levels" (l. 223 and abstract l. 31), i.e., to full depth?
WOA13 input data have monthly resolution down to 500 m for the nutrients, and down to 1500 m for T, S, and $O_2$. WOA13 input data have quarterly-resolved files down to 500 m for the nutrients, and down to full 5500 m depth for T, S, and $O_2$. Finally, WOA13 has annual mean files down to full 5500 m depth for nutrients, T, S, and $O_2$.
The quarterly-resolved fields of T, S, and $O_2$ show in some parts strong differences at high depths, e.g., at 3000 m (see attached figure 1, left column). The authors appear to interprete that as seasonality ("The seasonal amplitude of $A_T$ is progressively reduced at depth" l. 405f/Figure S7, and l. 310-315)?
For perspective, WOA13 quarterly variations at 3000 m depth have (e.g., in the Southern Ocean) a range up to 0.4 °C (on a total range of variability of ca. 40 °C), 0.05 psu (on a total range of ca. 2 psu), and beyond 40 µmol/kg (on a total range of ca. 400 µmol/kg). This certainly exceeds any expectation for a seasonal cycle, e.g., of oxygen, and demonstrates rather a data coverage and/or mapping issue. (Please consult / check the data coverage fields 'x dd'!)

This has three consequences:

1. The authors must decide until what depth they use which monthly-/ quarterly-/ annually-resolved WOA13 input fields, determining until what depth they can claim to provide a monthly-/ quarterly-/ annually-resolved $A_T$ climatology.

This must be 500 m, if they decide that their seasonality is caused by the organic matter cycle, reflected through both oxygen and nutrients variations (l. 319/321) (summed 57 % relative importance). It may be 1500 m if they decide that monthly-resolved oxygen (16 % relative importance), together with annual means of the nutrients (summed 41 % relative importance), may provide a sufficient driver for $A_T$ seasonality to the NNGv2 network, but this already must be clearly justified. It would be quite a stretch for any reasonable seasonality below 1500 m, and I would suggest to revert to annual mean WOA13 fields,

rather than the quarterly ones, to built the $A_T$ climatology. Please consult the 'x dd' data coverage fields, too, to assess if sufficient data went into the monthly / quarterly fields for a robust seasonality – or a robust assessment at all.

2. The seasonality and its limits must be made transparent through the author's work/manuscript. It should not be the task of the reviewer / user to check.

3. The CANYON-B/WOA13 comparison (l. 310-315) must be moved to the Climatology section 3.4 rather than the Neural network analysis 3.1 and discussed accordingly. A computation method "using relatively few input variables (position, time, temperature, salinity and oxygen)" (l. 305) is more prone to bad input data in one variable than a method that uses all the variables as inputs (l. 309). Particularly, (1) if the only biogeochemical predictor may be biased (oxygen; CANYON-B) rather than just one out of a total of three (LIARv2) or four (NNGv2), of which the three nutrients nitrate, silicate, and phosphate just have a WOA13 mean annual field below 500 m (!); (2) if exactly this predictor has a "seasonal" $O_2$ amplitude at 3000 m of up to 40 µmol/kg (versus the time / decimal year with a pretty modest variability of ca. 0.5 years on a total range of 40 years!); and (3) considering the strong correspondence at 3000 m between "seasonal" oxygen and "seasonal" $A_T$ amplitude (attached figure 1 right column).

To claim / blame the time for these variations in $A_T$ is pretty bold and wrong.[1]

Please correct the text, if you decide to keep it, and please make an effort for a balanced comparison of methods!

Please also correct the conclusion (l. 471/472), which neglects the impact of the quality of the WOA13 seasonality.

[Figure]

Figure 1: Left column: WOA13 "seasonal" amplitude at 3000 m of temperature (top), salinity (middle), and oxygen fields (bottom) from WOA13 quarterly- resolved files, which are (probably?) used as input to the $A_T$ climatology and for CANYON-B 'comparison' (l. 310-315). Right column: $A_T$ "seasonal" amplitude at 3000 m for the author's monthly climatology (probably?) based on WOA13 quarterly fields of T, S, $O_2$ and annual mean fields of nutrients as input (top), LIARv2 using WOA13 quarterly fields of T, S, $O_2$ and annual mean fields of nutrients as input (middle), and CANYON-B using just WOA13 quarterly fields of T, S, and $O_2$ (bottom). Note the correspondance of elevated patches of "seasonal" amplitude between WOA13 quarterly $O_2$ and CANYON-B $A_T$ in the Southern Ocean (a.k.a. 'garbage in, garbage out').

**AR**: We have taken in consideration your suggestion of reverting to an annual climatology below 1500m. On the other hand, we have created new monthly climatologies of nutrients up to 1500m using CAYON-B over the monthly fields of temperature, salinity and oxygen (with a modification, see Appendix A in the manuscript) from WOA13. In addition to the coherence reason to have the same time resolution in all the predictor variables, we have seen that the variability of these new climatologies is in general more accurate than that of WOA13 ones when the unique locations to create a monthly climatology from measured data (that is, time-series) are assessed (Appendix A). The neural network approach of CANYON-B seems to be a more accurate method to create climatologies of nutrients than that used by WOA13 (Appendix A). Therefore, we have specified the limits of the seasonality in the new manuscript as well as the inclusion of the new climatologies of oxygen and nutrients in the supplementary material that were used to create the new $A_T$ climatology.

**1.2 Uncertainty of climatology**

**RC**: Why is the monthly $A_T$ climatology not compared to the measured GLODAPv2 $A_T$ data? This would give a much more robust assessment than just two time series sites (HOT and BATS), at which the WOA13 input data arguably should be on the better side of the spectrum of possibilities, i.e., underestimating the climatology's uncertainty.

I don't see why such a comparison should be limited to locations with repeated sampling (l. 415) and not extended to times/months with repeated sampling (read: basin-crossing cruises as in GLODAPv2).

At the end of the day, the $A_T$ climatology should represent both temporal and spatial variability within its resolution – What better dataset to assess temporal and spatial variability than the largest available one, GLODAPv2? This should also include spatial / regional differences in the uncertainty.

At least some assessment of $A_T$ climatology uncertainty must be given before the dataset is acceptable for publication.

**AR**: The climatological data have to be compared to climatological data. The unique locations where a climatological value of $A_T$ can be obtained from measured data are time-series stations. Obviously, we did not have made the comparison with the two largest time-series to extract an uncertainty for all the climatology. If we had compared the climatological $A_T$ data to GLODAPv2 data, the differences would not be uncertainties otherwise would be more like an anomaly because that comparison is between a mean value over many years and a punctual value of $A_T$ (e.g., differences between black line and the limits of the shadow area in Fig. 11 and Fig. S8). Even for the repeated cruises in GLODAPv2 were the OVIDE section is the most repeated (7 times in the same month), there are not enough data available to generate a climatology with measured data and consequently to obtain an uncertainty in different areas of the ocean as in the time-series. Maybe that is the reason why the previous two monthly climatologies (Lee et al., 2006 and Takahashi et al., 2014) did not calculate an uncertainty and only gave an error analysis by zones which we have included here. Furthermore, as it is stated in the manuscript, WOA13 does not give an uncertainty of its objectively analyzed climatologies to derived one from the $A_T$ climatology.

**1.3 Subsurface layer hypothesis**

**RC**: Quoting from the text, the " winter relationship between inputs and $A_T$ needed to produce an all-season surface climatology are mostly preserved in [the] sub- surface layer." (l. 214).

However, the authors try to reinforce the hypothesis (1) using the nowinter network on all depths rather than just the surface, and (2) the NNGv2 network is never evaluated on the same data as the nowinter network. In fact, the numbers in table 7 might be nice to show, but don't give any indication about the validity of the subsurface layer hypothesis. The lower winter RMSE may just be related to less variability (at all depths?) during this one season compared to the three nowinter seasons. Many other rea- sons are plausible, too.

What the authors should do: (1) As the authors suggest and do in l. 370, control that the nowinter network is comparable to NNGv2 on the domain it is trained on by providing statistics for NNGv2 and nowinter network on the GLODAPv2 nowinter dataset (full depth and surface; can be moved to the supplementary if desired); (2) Provide statistics for NNGv2 and nowinter net- work on the GLODAPv2 winter dataset using only surface data (above the subsurface layer defined in lines 358-362). Only if they are comparable, or at least not exceedingly higher for nowinter over NNGv2, or not exceedingly higher than surface RMSEs in other seasons (e.g., GLODAPv2 nowinter dataset sur- face only), this would reinforce the subsurface layer hypothesis. (That is, the exclusion of the (scarce) winter data did not degrade the winter surface predic- tions (← nowinter network and NNGv2 network on GLODAPv2 winter surface data) thanks to the still present signature in the spring subsurface layer (← GLODAPv2 nowinter training data and full GLODAPv2 training data).)

Other than that, the subsurface layer hypothesis remains a hypothesis, which I'd doubt the NNGv2 to recognize and would suggest to remove.

Figure 8: Same question as before: Why are calculated NNGv2 $A_T$ values not shown? They should.

**AR**: We have added the statistics in different depth layers and showed them by network (NNGv2 and NNGv2_nowinter) and by dataset (winter and no_winter). It is clearly showed how the network trained without winter data (NNGv2_nowinter) is able to compute $A_T$ with non-significant differences compared to NNGv2 in any depth layer and dataset. The highest difference is in the 0-50m layer in the winter dataset where NNGv2 computes $A_T$ with one less unit RMSE than NNGv2_nowinter. Therefore, the subsurface layer hypothesis is reinforced as these good results show.

**2 Major points**

**RC**: Table 2/3: Why are LIARv2 and CANYON-B not added here? They should be added!
**AR**: Added.

**RC**: l.295. "The newest methods in the $A_T$ computation (...) model the GLODAPv2 $A_T$ with higher errors than the NNGv2 (Table 4)." This is because both LIARv2 and CANYON-B used only the GLODAPv2 $A_T$ subset for training where the 2nd QC was done, whereas our GLODAPv2 $A_T$ data for training included samples, too, where the 2nd QC was not done. "An analysis in a GLODAPv2 subset excluding the samples where the 2nd QC was not done for $A_T$ shows a reduction of the error [...]". NNGv2 results are not comparing to independent data in the GLODAPv2 no secondary QC subset because of correlations within cruises and the random splitting of cruises between testing/training, whereas LIARv2 / CANYON-B truly haven't seen any of these data.
**AR**: We have changed the neural network. Taking advantage of the publication of the new version of GLODAPv2 in ESSDD (Olsen et al., 2019), we have trained a new network using only secondary QC data, except for the Mediterranean Sea, to have a more coherent comparison between the three studies as you seem to suggest.

**RC**: Table 5 and 6 should be merged, such as table 4.
**AR**: Done.

**RC**: I still find it hard to justify the NNGv2 3RMSE network. The only clue to this one is that you remove an area (Arctic Ocean) with higher-than- average variability and, naturally, get better statistics. If you remove the same area from the NNGv2 assessment, you get the same, better statistics, too (table 1)!

There is still a lack of justification for the NNGv2 3RMSE, and no, I don't think that a few decimal places better RMSEs in a few out of the regions in table 3 justify the 3RMSE removal – you can get the same better performance here or there by having a closer look at the 10 NNGv2 networks you trained! Also, the conclusions from the authors response are not supporting their argument and are not substantial. ("[...] In this case, it is clear that omitting certain data causes a large difference between the networks." I don't see a large difference. If you insist, please use an appropriate test to verify significance; the "improvements in almost all of the zones suggest that they are because of this data deletion instead than the different local minimum reached in the error function." That's only what the authors want to see, I'd still see a different local minimum as more plausible. And no further evidence is given that this may not be the case.)

Please improve the NNGv2 vs. NNGv2 3RMSE network aspect or remove either one of the two.

**AR**: Following your suggestion we have considered to remove the NNGv2_3RMSE.

**RC**: What about the seasonal amplitude of $A_T$ at the time series sites of measured $A_T$ vs. NNGv2-measured inputs-based vs. NNGv2-WOA13-based?

**AR**: That comparison was carried out to show how the network can compute an accurate climatological value if the inputs are good, and also to show the differences between the input climatologies and the climatological values of the input variables obtained from measured data. However, we have deleted that line from the graph although we have decided to keep the discussion.

**RC**: To be complete, the subpolar North Atlantic should still be added to the current manuscript as test region for the current methods, even if it was the object of a previous work (Vazquez-Rodríguez et al., 2012).

**AR**: We did not add this area since we believe it would be repeating the same thing that is already done by Vazquez-Rodríguez et al. (2012). Furthermore, the analysis performed with the NNGv2_nowinter tests multiple regions around the world showing good results.

**3 Minor points**

**RC**: l. 316: "The NNGv2 seems to associate the $A_T$ variability to the predictor variables in coherence with the processes that contribute to it." So, does it? Please give evidence or remove/rephrase.

**AR**: We have rephrase emphasizing the qualitative character of this discussion.

**RC**: Table S1: What does 'HS' mean?

**AR**: latitudes lower than 0º. We have changed it to <0º.

**RC**: Table S2/S3: Column headings 'relative ... lat>60° N' should probably correspond?

**AR**: Percentages between the number of samples in the specified range (column1) and the total number of samples with residuals beyond 3RMSE (sum of column 2 in Table S2. In table S3 that total amount of data is not shown).

**RC**: l. 30: missing subscript $A_T$

**AR**: Thank you, changed.

[revised manuscript text omitted]